



# Ice microphysical processes in the dendritic growth layer: A statistical analysis combining multi-frequency and polarimetric Doppler cloud radar observations

Leonie von Terzi[1], José Dias Neto[2], Davide Ori[1], Alexander Myagkov[3], and Stefan Kneifel[1]

[1]Institute of Geophysics and Meteorology, University of Cologne, Cologne, Germany
[2]Delft University of Technology, Department of Geosciences and Remote Sensing, Delft, Netherlands
[3]Radiometer Physics GmbH, Meckenheim, Germany

**Correspondence:** Leonie von Terzi (lterzi@uni-koeln.de)

**Abstract.** The dendritic growth layer (DGL), defined as the temperature region between –20 and –10 ° C, plays an important role for ice depositional growth, aggregation, and likely also secondary ice processes. The DGL has been found in the past to exhibit specific observational signatures in polarimetric and vertically pointing radar observations. However, consistent conclusions about their physical interpretation have often not been reached.

5      In this study, we exploit a unique three months dataset of mid-latitude winter clouds observed with vertically pointing triple-frequency (X-, Ka-, W-Band) and polarimetric W-Band Doppler radars. In addition to standard radar moments, we also analyse the multi-wavelength and polarimetric Doppler spectra. New variables, such as the maximum of the spectral ZDR ($sZDR_{max}$), allows us to analyse the ZDR signal of asymmetric ice particles independent of the presence of low-ZDR producing aggregates. This unique dataset enables us to investigate correlations between enhanced aggregation and evolution of small ice particles in the DGL. For this, the multi-frequency observations are used to classify all profiles according to their maximum average aggregate size within the DGL. The strong correlation between aggregate class and KDP confirms the expected link between ice particle concentration and aggregation. Interestingly, no correlation between aggregation class and $sZDR_{max}$ is visible. This indicates that aggregation is rather independent of the aspect ratio and density of ice crystals. A distinct reduction of mean Doppler velocity in the DGL is found to be strongest for cases with largest aggregate sizes. Analyses of spectral edge velocities suggest that the reduction is the combined result of the formation of new ice particles with low fall velocity and a weak updraft. It appears most likely that this updraft is the result of latent heat released by enhanced depositional growth. Clearly, the strongest correlations of aggregate class with other variables are found inside the DGL. Surprisingly, no correlation between aggregate class and concentration or aspect ratio of particles falling from above into the DGL could be found. Only a weak correlation between the mean particle size falling into the DGL and maximum aggregate size within the DGL is apparent. In addition to the correlation analysis, the dataset also allows to study the evolution of radar variables as a function of temperature. We find the ice particle concentration continuously increasing from –18 ° C towards the bottom of the DGL. Aggregation increases more rapidly from –15 ° C towards warmer temperatures. Surprisingly, KDP and $sZDR_{max}$ are not reduced by the intensifying aggregation below –15 ° C but rather reach their maximum values in the lower half of the DGL. Also below the DGL, KDP





and sZDR$_{max}$ remain enhanced until –4 ° C. Only there, additional aggregation appears to deplete ice crystals and therefore
reduce KDP and sZDR$_{max}$.

The simultaneous increase of aggregation and particle concentration inside the DGL necessitates a source mechanism for
new ice crystals. As primary ice nucleation is expected to decrease towards warmer temperatures, secondary ice processes
are a likely explanation for the increase in ice particle concentration. Previous laboratory experiments strongly point towards
ice collisional fragmentation as a possible mechanism for new particle generation. The presence of an updraft in the tem-
perature region of maximum depositional growth might also suggest an important positive feedback mechanism between ice
microphysics and dynamics which might further enhance ice particle growth in the DGL.

## 1 Introduction

Recent space-borne analyses underline once more the importance of understanding ice growth processes in clouds as more
then 70% of global precipitation is found to be generated via the ice phase (Heymsfield et al., 2020; Field and Heymsfield,
2015; Mülmenstädt et al., 2015). The Dendritic Growth Layer (DGL), usually located between –20 and –10 ° C, is known to
play an important role for the growth and evolution of ice and snow in clouds. The reasons for its importance are manifold:
The difference in saturation vapour pressure between ice and liquid reaches a maximum at –12 ° C which favours depositional
growth by the Wegener-Bergeron-Findeisen (WBF) process (Korolev, 2007). In addition, the particular plate-like shapes that
particles grow into in the DGL lead to a distinct maximum in the depositional growth rate at –15 ° C, where dendritic particles
can exceed 1.5 mm in size within 10 minutes growth time at liquid water saturation (Takahashi, 2014). The fragile structure
of those particles has also been found by airborne in-situ observations (Schwarzenboeck et al., 2009; Hobbs and Rangno,
1998, 1990, 1985) and laboratory experiments (Takahashi et al., 1995; Griggs and Choularton, 1986; Vardiman, 1978) to favour
collisional breakup. This secondary ice process (SIP) gains increasing attention by the scientific community (e.g., Georgakaki
et al., 2022; Phillips et al., 2018) as an important process which could explain the discrepancy between number of ice nucleating
particles (INP) and ice particle number concentration (IPNC) (Kanji et al., 2017). Unlike for example the Hallett-Mossop rime
splintering process (Field et al., 2017; Hallett and Mossop, 1974), ice collisional fragmentation could provide new secondary
ice particles over a wide temperature range. Finally, ice particles which are multiplied in number by SIP and grow rapidly in
the DGL have been found to also aggregate very efficiently in the DGL (Lamb and Verlinde, 2011). Especially the branched
structure of dendrites appears to be responsible for enhancing the stickiness of the crystals and thus favouring the formation
of aggregates (Connolly et al., 2012). The various ice growth signatures in the DGL which can be observed for example with
radars could be linked to the intensity of surface precipitation (Trömel et al., 2019, and references therein). This highlights
again the importance to properly understand the interplay of microphysical processes in the DGL in order to accurately model
surface precipitation.

Dual-polarization radar observations are a powerful tool to observe the result of several of the aforementioned growth pro-
cesses in the DGL. The abundance of plate-like particles in the DGL leads to vertically distinct layers of enhanced differential
reflectivity (ZDR) and propagational differential phase shift (KDP) close to the –15 ° C region (Griffin et al., 2018; Schrom





and Kumjian, 2016; Schrom et al., 2015; Moisseev et al., 2015; Thompson et al., 2014; Bechini et al., 2013; Andrić et al., 2013; Kennedy and Rutledge, 2011; Trapp et al., 2001, among others). Interestingly, the ZDR layer appears at slightly higher altitudes as the KDP enhancement. ZDR is independent of the particle concentration but increases strongly with particle den-

sity and aspect ratio (Kumjian, 2013). However, the formation of aggregates leads to a decrease of ZDR despite the presence of asymmetric crystals which explains its layered structure. KDP is strongly related to particle concentration and, in contrast to ZDR, is not reflectivity-weighted, and thus not strongly influenced by the presence of large aggregates. While the general connection of ZDR and KDP layers and intensive plate-like growth and subsequent aggregation is widely accepted, a definite conclusion on the reasons for the distinct vertical structure has not yet been reached (e.g., Schrom and Kumjian, 2016; Schrom

et al., 2015; Moisseev et al., 2015).

Polarimetric information from the DGL is mostly lacking in vertically pointing radar observations. Similar as slant measuring radars, vertically pointing radars commonly observe a rapid increase in the radar reflectivity factor (Ze, henceforth called reflectivity) in the DGL, in particular at –15 ° C (e.g., Schrom and Kumjian, 2016; Zawadzki, 2013). The Doppler spectra collected with zenith-pointing radars revealed two distinct features in the DGL: First, the Doppler spectra often reveal an

additional slow secondary mode in the DGL. In case of low turbulence and weak vertical air motions, the Doppler velocities can be related to the particles' terminal fall velocity. Apparently, the second spectral mode indicates the formation of new ice particles that often increase in fall speed and eventually merge with the main aggregate mode falling from higher altitudes. Second, the mean Doppler velocity (MDV) often reveals a slight but temporally very persistent reduction in the DGL. Various explanations for those features have been presented in the literature (see for example discussion presented in Schrom and

Kumjian (2016); Schrom et al. (2015)). Zawadzki (2013) argue that vertical air motion at –15 ° C is necessary to enhance supersaturation which enables the nucleation and subsequent growth of plate-like particles. Other authors assign the MDV reduction simply to the evolution of a new secondary mode in the spectrum (e.g., Oue et al., 2018; Moisseev et al., 2015, 2009; Shupe et al., 2004; Zawadzki et al., 2001; Field, 2000, and references therein). The explanations for the origin of a new, slow ice particle mode include sedimentation of ice particles into the DGL from higher altitudes (Moisseev et al., 2015), enhanced

primary nucleation due to upward air motion (Zawadzki, 2013), as well as secondary ice particle formation (e.g., Kennedy and Rutledge, 2011). Also buoyancy-driven upward motion due to latent heat release of rapidly growing ice particles by water vapour deposition has been discussed as a potential reason for the decrease of MDV (Schrom and Kumjian, 2016).

An increasing number of ground-based sites are equipped with polarimetric and multi-frequency cloud radars. The use of higher frequencies does not only substantially increase the backscattered signal, especially of small ice particles scattering in the

Rayleigh regime, but it also reduces observational limitations in some polarimetric variables. KDP is the range derivative of the differential phase shift and generally affected by high measurement noise. As KDP is inversely proportional to the wavelength, it can be more reliably estimated at shorter wavelength, also for small concentrations of asymmetric ice particles (Bringi et al., 2001). Moreover, even particles whose reflectivity values are below the radar detection level will cause some differential phase shift. Hence, KDP is sensitive also to the presence of extremely small, asymmetric ice crystals such as those expected to be

produced by SIP. The elevation dependence of polarimetric cloud radar observations allows to infer shape, orientation, and apparent density of ice crytsals (e.g., Myagkov et al., 2015; Matrosov et al., 2012). In addition, most polarimetric cloud radars



provide also polarimetric Doppler spectra which allow to assign the polarimetric signatures to specific Doppler velocities. Most previous studies used LDR spectra from vertically pointing radar observations to investigate the evolution of columnar and needle particles between –10 and 0 ° C (e.g., Giangrande et al., 2016; Oue et al., 2015). Spectral ZDR from an S-Band

radar system has been used by Spek et al. (2008) to retrieve particle size distribution of aggregates and plates. Pfitzenmaier et al. (2018) analysed spectral LDR from a zenith pointing Ka-Band and spectral ZDR from a slant-viewing S-Band radar to study ice particle growth along fall streaks.

Aggregation in the DGL can be only indirectly detected by radar polarimetry as a reduction of for example ZDR and concurrent increase of radar reflectivity (Kumjian, 2013). In contrast, the increase in mean particle size can be well observed

as an increasing reflectivity difference in multi-frequency cloud radar observations (e.g., Kneifel et al., 2011; Liao et al., 2008, 2005; Matrosov, 1992). Ice particles increasing in size begin to scatter less radiation back relative to particles that can be still approximated by Rayleigh scattering (usually valid if particles size » wavelength). As this deviation happens first at the shorter wavelength, the logarithmic reflectivity difference (also called dual wavelength ratio, DWR) increases with the mean size of the particle size distribution (PSD). Also DWR can be resolved spectrally, which allows to constrain particle scattering

models (Kneifel et al., 2016), to retrieve the particle size distribution (Mróz et al., 2021; Barrett et al., 2019), or to separate attenuation and differential scattering effects (e.g., Li and Moisseev, 2019; Tridon and Battaglia, 2015).

The majority of previous radar studies on the DGL focus their analysis on a number of case studies. A more statistical investigation is presented by Trömel et al. (2019), where X-Band radar observations of 52 stratiform precipitation cases obtained close to Bonn, Germany, were analysed using quasi-vertical profiles (QVP, Ryzhkov et al., 2016). They found a correlation of

KDP and Ze in the DGL and were able to link signatures in the DGL to surface precipitation. Similarly, in a statistical analysis of 27 days of C-Band observations close to the city of Turin, northern Italy, Bechini et al. (2013) linked enhanced KDP in the DGL to an enhanced Ze at the surface. Schneebeli et al. (2013) analysed a dataset of polarimetric X-Band radar observations of clouds ranging in temperature between -30 to 0 ° C collected in the Swiss Alps. Interestingly, they were unable to find a distinct KDP maximum in the DGL but rather a continuous increase of KDP and Ze towards warmer temperatures related to a

general increase of the ice water content (IWC).

Only a few studies attempted to combine different radar approaches for studying the DGL. Oue et al. (2018) used vertically pointing and slant-viewing polarimetric cloud radars to study the DGL in Arctic clouds. By combining Doppler spectra of a vertically pointing Ka-Band radar with slant polarimetric observations, they were able to assign the increasing ZDR signatures in the DGL with the slow, secondary mode in the reflectivity Doppler spectra. A similar correlation of spectral bi-modalities

and polarimetric signatures in the DGL have also been identified in mid-latitude clouds (Moisseev et al., 2015).

In this study, we present an in-depth analysis of vertically pointing triple-frequency (X-, Ka-, W-Band) Doppler spectra combined with spectral polarimetric observations from a W-Band cloud radar operated at a fixed 30° elevation angle. To our knowledge, such a combined multi-frequency analysis including spectral polarimetric observations obtained at W-Band with a simultaneous transmit simultaneous receive (STSR) mode radar have not been presented so far. The 3-months dataset of

winter clouds observed at a mid-latitude European site close to Cologne, Germany, are described in Section 2.1. A combined view with the various radar observables on the DGL is illustrated for a case study in Section 3. The case study description





provides an overview of typical radar signatures which have been observed in the DGL in previous studies as well as an introduction to new observables based on spectral polarimetry and multi-frequency observations. In Section 4 a statistical analysis is presented, which aims to connect polarimetric and spectral signatures dominated by newly generated ice crystals to the maximum aggregate size reached in the DGL. The section also includes a spectral analysis aimed to disentangle the contributions of upward air motion and secondary ice particle mode on the MDV reduction observed in the DGL. In Section 5.1 we summarize the vertical evolution of the various radar variables with a special focus on the temperature level where changes in the different variables are most pronounced. Profiles from laboratory experiments are added to this conceptual picture to allow an in-depth discussion of the most likely evolution of microphysical processes in the DGL. In Section 5.2 the role of sedimenting particles from higher altitudes and especially cloud top temperature on the signatures in the DGL are shortly discussed. The main findings of our statistical analysis are summarized in Section 6.

## 2 Data and Methods

### 2.1 TRIPEx-pol campaign

The results presented in this study are based on a multi-month dataset obtained during the campaign "TRIple-frequency and Polarimetric radar Experiment for improving process observation of winter precipitation" (TRIPEx-pol). The campaign took place from November 2018 until January 2019 at the Jülich ObservatorY for Cloud Evolution Core Facility (JOYCE-CF, Löhnert et al. (2015), 50°54′31″ N, 6°24′49″ E; 111 m above mean sea level) located ca. 40 km west of Cologne, Germany. Similar to a previous winter campaign (TRIPEx, Dias Neto et al., 2019), radar Doppler spectra and moments were continuously collected from a combination of vertically pointing triple-frequency (X-, Ka-, and W-Band) radars. The main difference to the earlier TRIPEx campaign is an extension of the observational capabilities by two additional radars: A new X-Band radar with better sensitivity and the possibility to record Doppler spectra as well as a scanning polarimetric Doppler W-Band radar (Table 1). The vertically pointing and the scanning W-Band radars are both frequency modulated continuous wave (FMCW) systems manufactured by Radiometer Physics GmbH (Myagkov et al., 2020; Küchler et al., 2017). The X- and Ka-Band systems are pulsed radar systems manufactured by Metek GmbH (Mróz et al., 2021; Görsdorf et al., 2015). All four radar systems were installed at the same roof platform within horizontal distances of less then 20 m. The resolution in range and time were adjusted to allow a very close radar volume matching (Table 1). The polarimetric W-Band radar was measuring at 30° constant elevation (CEL) for intervals of 5 minutes towards West. In between the CEL measurements, the radar was performing single range height indicator (RHI, from 30° to 150° elevation) and plan position indicator (PPI, at 85° elevation) scans intended for wind profiling. Auxiliary instruments at JOYCE including rain gauges, microwave radiometers, and Doppler wind lidars provide additional information about the atmospheric state and precipitation on the surface (for further details see Löhnert et al. (2015)). The combination of various remote sensing instruments also allows the continuous generation of Cloudnet classification and categorisation products (Illingworth et al., 2007). Besides in-situ and remote sensing observations, Cloudnet products also incorporate thermodynamic and wind information for JOYCE-CF extracted from analysis fields provided by the European Centre for Medium-Range Weather Forecasts (ECMWF) Integrated Forecast System (IFS) model.





**Table 1.** Technical specifications of the four radars that were deployed during the TRIPEx-pol campaign. The W-Band and the W-Band pol radar are FMCW radars, therefore the range resolution, Doppler velocity resolution and the Nyquist range vary for the different chirps. The values in this table are valid for the lowest chirp region (W-Band: 215-1475 m, W-Band pol: 107-715 m). The full chirp tables for both W-Band radars are provided in Appendix A.

| Specifications | X-Band | Ka-Band | W-Band | W-Band pol |
|---|---|---|---|---|
| Frequency [GHz] | 9.4 | 35.5 | 94.1 | 94.0 |
| Polarimetry | single-pol | LDR | LDR | STSR |
| Number of spectral averages | 10 | 20 | 13 | 28 |
| Half-power beam width [°] | 1.0 | 0.6 | 0.6 | 0.6 |
| Range resolution [m] | 36.0 | 36.0 | 36.0 | 35.8 |
| Temporal resolution [s] | 2 | 2 | 3 | 7 |
| Sensitivity at 1 km [dBz], 2s integration time | −50 | −63 | −58 | −58 |
| Maximum range [km] | 12 | 15 | 16 | 16 |
| Doppler velocity resolution [m s$^{-1}$] | 0.038 | 0.025 | 0.04 | 0.05 |
| Nyquvist range [m s$^{-1}$] | ± 78 | ± 20 | ± 10 | ± 6 |

## 2.2 Processing of the vertically pointing radar data

The dataset from the three vertically pointing radars was processed in three levels. Level 0 contains the regridded and cleaned Doppler spectra. Level 1 contains the radar moments calculated from the Level 0 dataset (Section 2.2.1). The level 2 processing follows closely the method presented in Dias Neto et al. (2019). It includes corrections for radar specific calibration offsets as well as gas, liquid, and ice attenuation (Sections 2.2.2 and 2.2.3). Key methods of the processing steps will be discussed in the following subsections.

### 2.2.1 Doppler spectra processing

Despite the similarity of the radar resolutions in space and time (see Table 1), the measured Doppler spectra of each radar had to be regridded to a reference time-height grid. For the reference grid, we chose a temporal resolution of 4 s and a range resolution of 36 m. The original data were matched to the reference grid using the method of nearest neighbours but only considering data points with a maximum displacement of ±17 m in range and ±2 s in time.

The center frequency of the vertically pointing W-Band radar had to be slightly changed from 94.00 GHz to 94.12 GHz in order to avoid interference with the W-Band polarimetric radar. This change caused some spectral artefacts which are caused by the spectral impurities of the used chirp generator. The level of impurities is considerably lower when the radar operates at the default center frequency. Also some weak "side lobes" appeared in the X-Band Doppler spectra when stronger signals were present close to the ground.



In order to identify the spectral region with "true" atmospheric signal, we selected the Ka-Band radar as reference. The Ka-Band radar provides the highest sensitivity of all radars and its Doppler spectra showed no artefacts. Due to the different heights of the lowest usable range gates of the radars, return signals below 400 m altitude were omitted. The Ka-Band Doppler

spectra were used to derive a spectral mask for each range gate and time step. For this, we first identified the spectral edges by subtracting the noise floor using the common method by Hildebrand and Sekhon (1974) and then locating the outermost spectral bins which exceed the noise level by 3 dBz. This spectral mask was then applied to filter the Doppler spectra of the other two vertically pointing radars. This filtering could unfortunately not remove all artefacts, as the W-Band artefacts did also sometimes overlap with the Doppler spectrum from real atmospheric targets.

From the regridded and filtered Doppler spectra the common radar moments are derived including equivalent radar reflectivity factor (Ze), mean Doppler velocity (MDV), spectrum width, and skewness. The Ka-Band spectra were also used to derive the fast falling edge and slow falling edge of the Doppler velocity of each spectrum. Those spectral edge velocities were derived in a similar way as the spectral mask. In case of strong atmospheric signals, spectral leakages might cause biases in the spectral edge velocity estimate. We mitigate this effect by neglecting all spectral lines which are lower than 40 dBz with respect to the

maximum spectral line. Examples of the derived spectral edge velocities are shown in Figure 3d where they are overlaid to the original spectra.

### 2.2.2 Evaluation of radar reflectivity calibration and antenna pointing

The reflectivity calibration of all four radars was evaluated using the drop size distributions (DSDs) measured during rainfall periods by the PARSIVEL optical disdrometer (Löffler-Mang and Joss, 2000) which was installed at JOYCE-CF in close

vicinity to the radars (Figure 1). The method is identical to the approach described in Dias Neto et al. (2019). The DSDs are used to calculate the Ze distribution for each radar frequency and rainfall event. This distribution is then compared to the measured Ze distribution at the lowest usable range gate (more details and discussion of uncertainties are provided in Dias Neto et al., 2019). The offsets estimated with this method for the three radars are 0 dBz for the X-Band, an underestimation of 3 dBz for Ka-Band, and an overestimation of 2 dBz for the W-Band. We applied the disdrometer-based method to 21 rainfall cases

and found no systematic temporal drifts of the estimated offsets. The W-Band radar data obtained during the TRIPEx-pol campaign has also been used in Myagkov et al. (2020) to evaluate different calibration methods including also disdrometer based methods. Our estimated bias of 2 dBz lies within their estimated values of 0.5 to 2.1 dBz for the vertically pointing W-Band radar. For the polarimetric W-Band radar they found an underestimation of $0.7\pm0.7$ dBz.

Accurate zenith pointing is crucial for the analysis of Doppler spectra and MDV in order to avoid velocity biases induced

by horizontal wind. The absolute pointing of the Ka-Band radar has been evaluated using sun-tracking scans (e.g., Muth et al., 2012). The pointing accuracy during the campaign was found to be better than $\pm$ 0.1° in elevation and azimuth. For the non-scanning X- and W-Band radar, the pointing could only be evaluated relative to the absolute calibrated Ka-Band radar. Following the approach shown in Kneifel et al. (2016), the pointing of the X- and W-Band radars has been evaluated in relation to the Ka-Band radar. For this, the difference in mean Doppler velocity (MDV) between X- and Ka-Band (Ka- and W-Band)

has been analysed in dependency of the horizontal wind speed and direction obtained from Cloudnet. The analysis of the MDV





differences obtained during TRIPEx-pol indicated that the misalignment between X- and Ka-Band as well as W- and Ka-Band did not exceed 0.1°.

### 2.2.3 Attenuation correction and relative DWR calibration

At cloud radar frequencies, atmospheric gases and hydrometeors cause attenuation which generally increases with frequency.
Similar to Dias Neto et al. (2019), we first corrected the Ze profiles for the estimated attenuation by gases. The gas attenuation profiles were calculated with the Passive and Active Microwave TRAnsfer model (PAMTRA, Mech et al., 2020) which takes into account contributions by nitrogen, oxygen and water vapour. Profiles of temperature, humidity and pressure provided by CloudNet were used as input for PAMTRA. Estimating the vertical profile of attenuation by liquid and frozen hydrometeors is challenging as accurate profiles of hydrometeor mass content and size distributions are required. As profile information
of liquid and ice are unavailable, we only estimate the total path integrated attenuation following the approach presented in Dias Neto et al. (2019).

This method leverages on the fact that small ice particles can be assumed to be Rayleigh scatterers for which the radar reflectivity factor

$$\text{Ze}(\lambda) = \frac{\lambda^4}{\pi^5 |K_\lambda|^2} \int \sigma_\lambda(D) N(D) \mathrm{d}D \tag{1}$$

is independent of the wavelength $\lambda$, if we assume a constant dielectric factor $|K_\lambda|^2$ (e.g., Kneifel et al., 2015; Hogan et al., 2000). $\sigma_\lambda(D)$ is the backscattering cross section of a particle with maximum size $D$ and $N(D)$ is the particle size distribution. The dual-wavelength ratio

$$\text{DWR}_{\lambda_1, \lambda_2} = \text{Ze}_{\lambda_1} - \text{Ze}_{\lambda_2} \tag{2}$$

is the difference in reflectivity in logarithmic units at two wavelengths $\lambda_1, \lambda_2$ (usually with $\lambda_1 > \lambda_2$). With the particle size
getting closer to the wavelength, the backscattering cross section increases less than expected from the Rayleigh approximation due to the destructive interference of electromagnetic waves scattered by various parts of the particle (see Figure C1a). This deviation from the Rayleigh scattering behaviour starts first at the smallest wavelength. As a result, increasing DWRs can be attributed to larger mean particle sizes in the radar volume. If the mean particle size becomes large enough, also the largest wavelength would transition to the non-Rayleigh regime and the DWR will increase slower, eventually reaching a saturation
point (Mason et al., 2019).

The total DWR measured under real conditions at a certain range can be written as the sum

$$\text{DWR}_{\lambda_1, \lambda_2} = \text{DWR}_{\text{scat}} + \text{DWR}_{\text{hard}} + \text{DWR}_{\text{att}}. \tag{3}$$

$\text{DWR}_{\text{scat}}$ is due to differential scattering of particles. Constant hardware related offset $\text{DWR}_{\text{hard}}$ might originate for example from radar miscalibration or differential radome attenuation. Differential attenuation causes a propagational component $\text{DWR}_{\text{att}}$
which accumulates with increasing range. The major contributions to total attenuation are due to rain, the melting layer, and





supercooled liquid water. Ice and snow also contribute to W-Band attenuation but as shown by Tridon et al. (2020) an ice water path larger then 1 kg m$^{-2}$ is needed to cause 1 dBz two-way attenuation.

As proposed originally by Hogan et al. (2000) and recently evaluated by Tridon et al. (2020), a reflectivity threshold can be used to identify cloud regions where DWR$_{scat}$ is negligible. The remaining DWR can then be attributed to the sum of
DWR$_{att}$ and DWR$_{hard}$. The X-Band profiles would be least affected by attenuation but the X-Band sensitivity is often too low to capture Rayleigh regions at high altitudes. Therefore, we use Ze profiles from the Ka-Band radar as our reference which have been corrected for gas attenuation. Ka-Band reflectivities between –30 dBz < Ze$_{Ka}$ < –10 dBz and between –15 dBz < Ze$_{Ka}$ < 0 dBz are used for estimating the non-scattering DWR components for W-Ka and X-Ka, respectively. To exclude partially melted particles, we additionally restricted the cloud regions used for the relative DWR calibration to be at least 1 km
above the melting layer. Following the approach in Dias Neto et al. (2019) we also exclude profiles where the number of valid measurements within a 15 minute time window is less then 300. Further, regions for which the variance of the DWRs exceeds 2 dB$^2$ or where the correlation of data points is less than 0.7 are not used. The estimated relative DWR offset for a moving time window of 15 minutes is then applied to the entire profile. As we expect the major contributions to the total attenuation from the rain part and the melting layer, this approach appears to be justified for the ice part of the cloud. In case of additional
attenuation in the ice part, for example due to layers of supercooled liquid, our approach would cause an overestimation of the true attenuation and hence our processed DWR$_{scat}$ values would underestimate the real DWR$_{scat}$ below the attenuation layer (even returning negative values). For convenience, in the following we use the radar Bands (X, Ka, W) instead of the wavelengths as indices (e.g. DWR$_{KaW}$). The DWR have also been derived spectrally (sDWR) by calculating the difference of the logarithmic power in each Doppler spectral bin. Identical corrections and relative calibrations as used for the DWR are
applied for sDWR.

### 2.3 Processing of the polarimetric radar data

The polarimetric W-Band radar observations were collected at constant 30° elevation and a fixed azimuth angle of 235° for 5 minute time periods. The azimuth direction is close to the main wind direction where most cloud systems have been advected from during the campaign (south-west to north-west according to wind information from Cloudnet). In order to minimize
time-lag differences related to different observation volumes of the slanted polarimetric radar and vertically pointed systems, we average the all measurements over 5 minutes.

The polarimetric radar moments as well as the polarimetric Doppler spectra have been projected to the height above ground and then regridded to the same reference grid as used for the vertically pointing observations. In order to reduce the noise of the specific differential phase shift (KDP), we first smoothed the differential phase shift $\phi_{dp}$ over 5 range gates (corresponding
to 180 m) using a moving window mean and then averaged it over 5 minutes before calculating KDP as half of the discrete range derivative of $\phi_{dp}$:

$$\text{KDP} = \frac{\Delta\phi_{dp}}{2 * \Delta r} \tag{4}$$

with $\Delta r$ being the distance between the range gates.





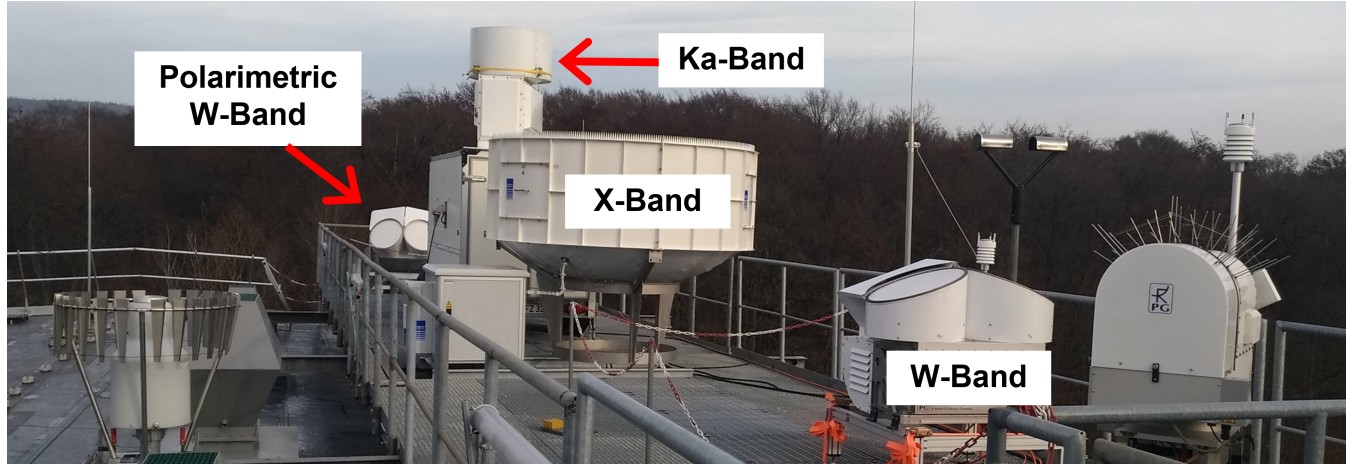

**Figure 1.** Measurement setup on the roof platform of JOYCE-CF in Jülich, western Germany (Löhnert et al., 2015). The four radars measuring during the TRIPEx-pol campaign were mounted in close vicinity to ensure optimal volume matching. A description of the radar setup can be found in Table 1 and Section 2. The radar measurement setup is complemented with a microwave radiometer HATPRO (in front of the W-Band radar at the far right of the picture), a PARSIVEL (behind the W-Band radar) and a PLUVIO precipitation gauge on the far left of the picture. Information about further instrumentation on the JOYCE-CF roof platform is given in Section 2 and in Löhnert et al. (2015).

The ZDR is defined as

$$\text{ZDR} = 10 \log_{10} \left( \frac{\text{Ze}_\text{H}}{\text{Ze}_\text{V}} \right) \tag{5}$$

with the radar reflectivity at horizontal polarization ($\text{Ze}_\text{H}$) and vertical polarization ($\text{Ze}_\text{V}$). The ZDR values are dominated by the particles contributing largest to the reflectivity. In addition to ZDR, we also derived the spectral ZDR (sZDR) as the logarithmic difference of horizontal and vertical power in each Doppler spectral bin. The maximum in sZDR called sZDR$_\text{max}$ indicates the presence of high ZDR producing ice particles within the radar volume even in cases where the ZDR is for example lowered by the presence of low ZDR producing aggregates. The quality of polarimetric measurements strongly depends on the signal-to-noise ratio (SNR). Variance in ZDR, $\phi_{dp}$, and $\rho_{hv}$ drastically increases at very low SNR (eq. 6.122 and section 6.5 in Bringi and Chandrasekar, 2001). Therefore, in order to use only high quality data for the following analysis, we omit all polarimetric observations when SNR (or in case of sZDR, spectral SNR in a spectral bin) is less then 10 dB.

### 2.4 Description of the dataset

Most clouds and precipitation events that occurred during the 89 days of the campaign were caused by mid-latitude frontal systems which are common during wintertime at JOYCE-CF. According to the Cloudnet classification, ice and mixed phase clouds were present 46.6 % of the time (1029 h). Rainfall was observed during 9.2 % (202.8 h) of the time causing a total accumulation of 152.7 mm of rainfall measured by the Pluvio weighing gauge installed at JOYCE-CF. On three days, snowfall was reaching the ground. A total of 18.6 h of snowfall produced a total liquid equivalent accumulation of 10.8 mm. The coldest





temperature of –7.0 ° C was observed on 24 January 2019, one of the three days with snowfall on the ground. The warmest
temperature, reaching 16.7 ° C, during the campaign was observed on 24th November 2018.

## 3   Snowfall case study: 30th January 2019

With the following snowfall case study we aim to provide an impression of the data quality and illustrate the complementary
information in spectral multi-frequency and polarimetric observations. Common observational features of the DGL visible in
standard radar moments as well as in new spectral variables will be discussed.

On 30th January 2019 a frontal system passed over JOYCE-CF and caused snowfall reaching the surface with a liquid
equivalent accumulation of 6.6 mm. During the entire day, temperatures at the surface remained below freezing, ranging
between –2.4 ° C and –0.4 ° C. The various radar observables from zenith pointing and slant polarimetric observations are
displayed in Figure 2. The cloud system produced snowfall reaching the ground mainly between 5 and 19 UTC. The overall
cloud structure observed in zenith is very similar to the radar measurements at 30° elevation. This similarity was regularly
observed for the three-months time period probably due to the predominantly large scale structure of the winter precipitation
observed during the campaign. The snowfall reaching the ground was mostly comprised of unrimed or only slightly rimed
crystals and aggregates. Visual observations between 9 and 10 UTC at the site revealed presence of stellar and dendritic
crystals reaching up to 4 mm in size mixed with unrimed aggregates with maximum sizes up to 10 mm (Figure B1). Also the
MDV (Figure 2b) throughout the case are found to be slower than –1.5 m s$^{-1}$ which indicates unrimed or only slightly rimed
particles (Kneifel and Moisseev, 2020). When plotting the DWR in the triple-frequency space (DWR$_{KaW}$ against DWR$_{XKa}$,
not shown), we also find a "hook shape" which has been previously found to indicate predominance of unrimed aggregates
(Kneifel et al., 2015).

The combined radar observations reveal several features which have been reported and discussed in previous literature
related to the DGL (Ori et al., 2020; Barrett et al., 2019; Griffin et al., 2018; Oue et al., 2018; Schrom et al., 2015; Moisseev
et al., 2015; Andrić et al., 2013; Bechini et al., 2013, among others). The Ze values (Figure 2a) rapidly increase at the –15 ° C
temperature level most likely due to an increase in the mean particle size as indicated by DWR$_{KaW}$ (Figure 2c). A layer of
enhanced sZDR$_{max}$ (Figure 2d) values up to 4 dB at –15 ° C indicates a rapid generation of asymmetric particles. According to
Myagkov et al. (2016), such values of differential reflectivity at these temperatures correspond to horizontally aligned strongly
oblate (plate-like) ice particles. For the following analysis, we define the aspect ratio similar to Takahashi et al. (1991) as the
ratio of the a- and c-axis of an ice crystal. Plate-like particles for example then have aspect ratios larger than 1. The decrease
of sZDR$_{max}$ from 4 to 2 dB towards lower layers indicates a change in particle properties (i.e. aspect ratios become closer to
unity and/or apparent ice density becomes smaller) of the strong ZDR producing particles found at –15 ° C. Also the KDP
(Figure 2e) shows an immediate increase at –15 ° C with values of up to 3 ° km$^{-1}$. Interestingly, both KDP and sZDR$_{max}$
remain enhanced down to the surface despite the ongoing aggregation indicated by increasing DWR$_{KaW}$ towards the ground.

Additional insights into the vertical evolution of particle populations and their contribution to radar moments can be gained
from the analysis of vertical profiles of Doppler spectra (Figure 3d-f). The first aggregates which produce a noticeable







**Figure 2.** Snowfall event occurring on 30 January 2019 at JOYCE. From the vertically pointing radars, the (a) Ze and (b) MDV at Ka-Band and the (c) $DWR_{KaW}$ are shown as time-height plots. From polarimetric observations at W-Band and 30° elevation angle (mapped to height above ground) the (d) maximum spectral ZDR $sZDR_{max}$ and (e) KDP are presented. In (a)-(e) the dashed red lines depict the -30 ° C and −15 ° C isotherms. Impressions of dendrites and aggregates sampled on the ground between 09:00 and 10:00 UTC are provided in Figure B1.

$sDWR_{KaW}$ signal (see Figure 3e)) appear at −18 ° C on the fast edge of the Doppler spectra (ca. −1 to −1.2 m s$^{-1}$). Interestingly, the spectral velocity where we find the largest sZe values at this temperature is at slightly lower velocities (−0.8 to −1 m s$^{-1}$). This could indicate that the number concentration of the aggregates producing the enhanced $sDWR_{KaW}$ region is still low. At around the same temperature level we also find a secondary mode in sZe and a broadening of $sDWR_{KaW}$ (indicated by magenta square in Figure 3d), e)). At its first appearance, the secondary mode shows initial spectral velocities close to



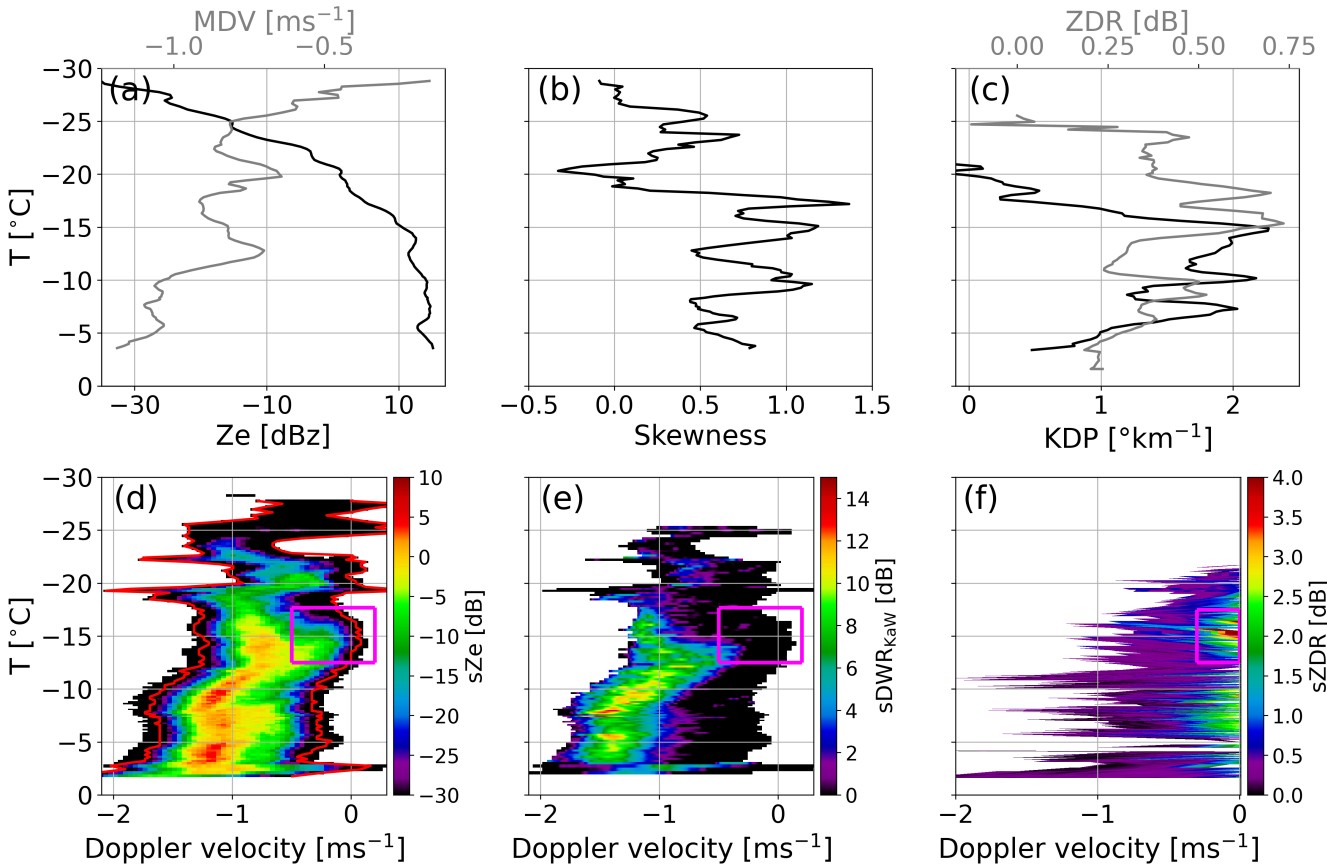

**Figure 3.** Radar profiles and spectrograms from the snowfall event occurring on 30 January 2019 at JOYCE at 13:30 UTC. Vertical profiles of (a) Ze and MDV, (b) skewness from the Ka-Band radar and (c) KDP and ZDR from the W-Band radar are shown. Spectrograms (i.e., vertical evolution of Doppler spectra) of (d) Ze from the Ka-Band are shown together with the derived spectral edge velocities (red lines). Panel (e) shows spectrograms of DWR$_{KaW}$ and (f) ZDR from the W-Band radar. The magenta squares in (d)-(f) depict the region where a secondary spectral mode is visible. An example of a single Doppler spectrum, where the second mode is visible, is also provided in Figure B2.

–0.3 m s$^{-1}$ which indicates small ice particles rather than super-cooled liquid droplets which typically produce a narrow spectral mode around 0 m s$^{-1}$ (e.g., Kalesse et al., 2016; Shupe et al., 2004). The spectral asymmetry introduced by the secondary

mode leads to a rapid change in the Doppler spectral skewness (Figure 3b). Starting at –18 ° C, the skewness increases from values close to unity (symmetrical spectra) to 1.4 at –17 ° C (Doppler spectra are skewed towards the slower falling side). The secondary mode appears alongside polarimetric signatures. KDP starts to increase at –20 ° C, reaching a maximum of 2.1 ° km$^{-1}$ at –15 ° C, correlating well with the increase and the maximum of sZDR of around 4 dB. The enhanced KDP and sZDR values indicate an increase in concentration and aspect ratio of small ice crystals.

For temperatures warmer than –15 ° C, the fall velocities and sZe (sDWR$_{KaW}$) of the secondary mode increase until they merge with the main mode at around –12 ° C. Interestingly, at this temperature we also see a distinct slow-down of the MDV





from approx. $-0.9$ m s$^{-1}$ at $-17\,°$ C to $-0.6$ m s$^{-1}$ at $-12\,°$ C. Looking at the spectrogram for this particular time, it appears as if the slow-down in MDV is the result of the new mode and a general shift of the entire spectrum to slower velocities. The later is compatible with the effect of upward air motion. This slow-down of the entire Ze spectrum is illustrated in Figure B2a)
which presents a zoomed view into the temperature region $-16\,°$ C to $-10\,°$ C. One can see that mainly the slow-down of the maximum of the main spectral mode at $-12.5\,°$ C leads to the slow-down in MDV.

Just slightly below the temperature level where the secondary mode merges with the main mode, we find the largest sDWR$_{KaW}$ values of up to 10 dB at spectral velocity bins between $-1$ and $-1.5$ m s$^{-1}$. At temperatures around $-10\,°$ C, the sZDR$_{max}$ values increase again roughly coinciding with the appearance of a weak secondary mode in the sZe and an in-
crease in KDP. The new particle mode as well as the enhanced sZDR$_{max}$ and KDP values mostly disappear at temperatures higher than $-5\,°$ C while sZe and sDWR$_{KaW}$ remain enhanced.

The signatures found in this case study are largely in agreement with radar signatures reported in previous studies about particle growth and aggregation in the DGL (Griffin et al., 2018; Schrom and Kumjian, 2016; Moisseev et al., 2015; Schrom et al., 2015; Andrić et al., 2013; Bechini et al., 2013, among others). However, also some differences are found, especially when
comparing our results to studies that analysed lower frequency polarimetric radar observations. Those observations frequently revealed layers of enhanced KDP and ZDR at the $-15\,°$ C level. One reason for the less layered appearance of KDP and ZDR in our case could be related to the higher frequency used for polarimetric observations in this study. As KDP is inversely proportional to the radar wavelength (e.g., Bringi et al., 2001), we are able to observe KDP signals from relatively small particle concentrations which are difficult to detect by low-frequency polarimetric radars. For example, a KDP signal of $1\,°$ km$^{-1}$
observed at W-Band would only be $0.1\,°$ km$^{-1}$ at X-Band. Also, the SNR is much higher for cloud radars since the maximum distance measured is smaller than for typical weather radars. As a result, only the maximum of the KDP enhancement close to the $-15\,°$ C level might be detectable by low frequency radars and regions with enhanced particle concentrations, but KDP values below the detection limit might be missed. A more detailed discussion of the expected similarities and differences of ZDR and KDP at X-Band and W-Band is provided in Appendix C.
In the following sections, we will apply our analysis to all ice containing clouds included in our dataset. As this case study illustrated, the combination of observations from the slant (Figure 2d-e) and the zenith direction (Figure 2a-c) appears to be reasonable, especially when applying an additional temporal averaging over 5 minutes to the profiles. This will allow us to link polarimetric and multi-frequency zenith observations in order to better understand which radar variables are connected to different intensities of aggregation in the DGL.

## 4   Profile classification by the mean aggregate size in the DGL using DWR$_{KaW}$

Aggregation becomes particularly active in the DGL causing rapid changes in radar quantities sensitive to the mean size, such as Ze or DWR (see also Figure 2). As a growing aggregate will deviate from Rayleigh scattering first at the shortest wavelength, we use DWR$_{KaW}$ as our most sensitive measure for the onset of aggregation. In order to exclude multi-layer or sublimation cases, we require the radar profiles to be continuous within the DGL. Following the approach presented in Dias Neto (2021),





**Table 2.** Definition of maximum $DWR_{KaW}$ intervals within the DGL (i.e., temperature region between –20 ° C and –10 ° C) used to classify radar profiles according to the particles' maximum mean mass diameters $D_0$ in the DGL. The last column denotes the number of available radar profiles with continuous observations within the DGL.

| max($DWR_{KaW}$) [dB] | class number | approx. $D_0$ [mm] | number of profiles |
|---|---|---|---|
| 0 - 1.5 | 1 | < 0.75 | 222 |
| 1.5 - 4.0 | 2 | 0.75 - 1.5 | 223 |
| 4.0 - 9.5 | 3 | 1.5 - 6 | 190 |

these profiles were sorted into three classes according to their maximum $DWR_{KaW}$ value reached within the DGL (Table 2). Assuming inverse exponential PSDs combined with particle and scattering properties of dendritic aggregates (Ori et al., 2021), we find the three $DWR_{KaW}$ classes representing mean mass diameters ($D_0$) ranging from 1 to 6 mm. The $DWR_{KaW}$ classes were chosen such that there is a similar number of profiles within each $DWR_{KaW}$ class. All profiles with $DWR_{KaW}$ values exceeding 9.5 dB are excluded as they are most likely related to partially rimed aggregates or due to an insufficient correction of W-Band
attenuation.

### 4.1   Relation of vertically pointing radar variables to the mean aggregate size in the DGL

After classifying all vertically pointing profiles according to their maximum $DWR_{KaW}$ in the DGL, we can now investigate how other radar moments evolve as a function of temperature for the different classes (Figure 4). The profiles of $DWR_{KaW}$ (Figure 4b) reveal that the maximum $D_0$ is reached at the lower end of the DGL (-10 ° C) with the strongest $DWR_{KaW}$ increase
found at temperatures warmer than –15 ° C. This is also the temperature, where we find $DWR_{XKa}$ (Figure 4c) to rapidly increase reaching values of 2 dB at –10 ° C for the highest $DWR_{KaW}$ class. Both DWRs only slightly change from –10 to –5 ° C. From –5 ° C towards the melting layer we find an additional increase in the DWRs, especially in $DWR_{XKa}$. This is in agreement with previous DWR studies (Ori et al., 2020; Dias Neto et al., 2019) and in-situ observations (Lawson et al., 1998). The most common explanation for this second aggregation layer is an increasing thickness of the quasi-liquid layer of the ice surface
causing the sticking efficiency to rapidly increase at T>–5 ° C coupled with enhanced depositional growth at around –5 ° C (Lamb and Verlinde, 2011).

Interestingly, small Ze differences between the aggregation classes are already visible at temperatures lower than –20 ° C (Figure 4a). At the top of the DGL, we find only a 3 dBz difference in Ka-Band reflectivities between the different aggregation classes which increase up to 10 dBz at the bottom of the DGL. Between –20 and –15 ° C, the slopes of the Ze profiles are
relatively similar. From –15 ° C to the bottom of the DGL, we find a relatively constant Ze for class 1, an unchanged linear increase for class 2, and a more rapidly increasing Ze curve for class 3. Unlike the DWRs, the Ze profiles remain relatively constant or even decrease for temperatures between –10 and 0 ° C. This behaviour might be related to non-Rayleigh scattering effects at Ka-Band (also indicated by increasing $DWR_{XKa}$) and/or a result of microphysics. For example, a simultaneous





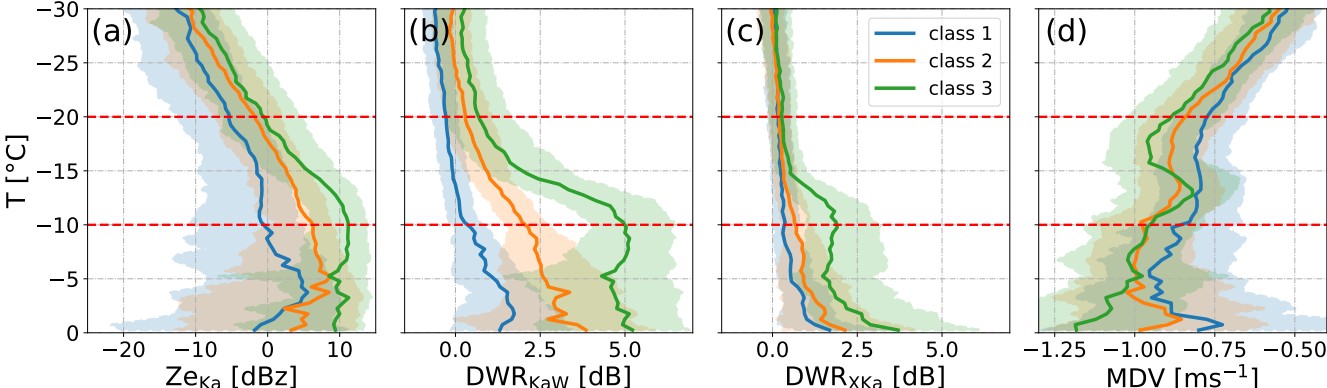

**Figure 4.** Median (solid line) and quantiles (shading) of (a) Ze at Ka-Band, (b) $DWR_{KaW}$, (c) $DWR_{XKa}$, and (d) MDV (Ka-Band) profiles stratified with temperature and classified into classes of maximum $DWR_{KaW}$ in the DGL (see class definition in Table 2). The temperature region of the DGL is indicated by the dashed red lines. Only profiles which are continuous in the DGL are considered.

increase in mean particle size and a decreasing number concentration caused by aggregation might have a compensating effect
on Ze.

The most intriguing signature, however, is found for the MDV in the DGL (Figure 4d). The MDV is increasing (in an absolute sense) from ca. –0.6 m s$^{-1}$ at –30 °C to –0.9 m s$^{-1}$ at –18 °C. This continuous increase is expected due to evolving particle size related to depositional growth and aggregation. Unlike Ze and $DWR_{KaW}$, the MDVs are only slightly larger for the larger $DWR_{KaW}$ classes. When the temperature increases above –16 °C, the largest $DWR_{KaW}$ class shows a pronounced
reduction of the MDV reaching a local minimum of 0.8 m s$^{-1}$ at –13 °C. At the bottom of the DGL, the MDV values increase to only slightly larger values (0.8 to 1.1 m s$^{-1}$) as compared to the top of the DGL. This "slow-down" of the MDV in the DGL appears to increase with $DWR_{KaW}$ class. Different explanations for this slow-down in the DGL have been discussed in literature. A common explanation for this effect is the occurrence of a new and slower ice mode in the Doppler spectrum (similar to the mode shown in Figure 2f) which would cause a reduction in the MDV (e.g., Schrom and Kumjian, 2016). An
alternative explanation proposes the existence of a frequently occurring upwind at –15 °C produced by large scale lifting (Zawadzki, 2013). A third explanation assumes that the latent heat released by enhanced depositional growth of ice particles in the DGL will locally increase buoyancy and eventually cause upward air motion (Schrom and Kumjian, 2016; Zawadzki, 2013). Obviously, if only the MDV is considered it is impossible to disentangle vertical air motion and microphysical effects. In the following section, we will extend the analysis to the Doppler spectra in order to better estimate the relative contributions
of microphysics and upward motion on the observed MDV slow-down in the DGL.

**4.2 Estimating the contribution of vertical air motion and new particle mode to the MDV reduction in the DGL**

The observed slow-down of the MDV in the DGL could be solely caused by the appearance of a slow, secondary particle mode or entirely due to an upward air motion. Both effects might even be connected to each other, for example if a large-scale updraft





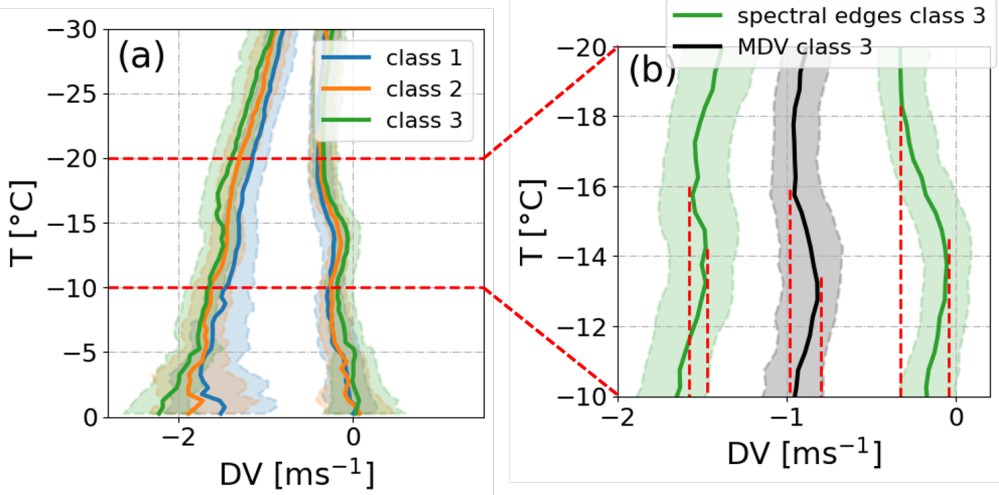

**Figure 5.** (a) Derived Doppler velocities of the spectral edges classified into $DWR_{KaW}$ classes (see also Table 2). Solid lines indicate the median, the shading denotes the quantiles of the distribution. The dashed red lines indicates the location of the DGL. (b) Zoomed view into the DGL region but only including the spectral edge velocities (green) and MDV (black) of class 3. The vertical dashed red lines in (b) visualize the points in the spectral edges and MDV which have been used to estimate the maximum velocity reduction (see details in the text).

is locally enhancing relative humidity which might then trigger nucleation of new ice particles. Alternatively, rapid depositional
growth of ice particles in the DGL might release latent heat which could cause a buoyancy driven upward air motion.

Vertical air motion can be derived from Doppler spectra if the terminal velocity of a spectral mode or distinct spectral feature is well known. In mixed-phase clouds, the presence of a narrow spectral peak due to small super-cooled liquid water droplets can sometimes be used to infer vertical air motion, assuming that the terminal velocity of the droplets is negligible (e.g., Zhu et al., 2021; Shupe et al., 2004). Unfortunately, a super-cooled liquid peak does not appear in all ice clouds and is often only
occurring in a relatively narrow height region of the cloud. However, vertical air motion is commonly assumed to impact all particles in the radar volume in the same way. As a result, the spectrum will be shifted to higher or lower Doppler velocities but without changing its shape. In contrast, a new particle mode will only affect the slow edge of the spectrum while the fast falling particles will be mostly unaffected. This slow-down effect due to new particle formation will be detectable as soon as the new mode overcomes the noise threshold even if no distinct secondary peak can be identified. Therefore, if the MDV slow-down is
solely an effect of vertical air motion, we expect the slow and the fast spectral edge to show an identical reduction in Doppler velocity. If the slow-down is caused by new particles only, the slow edge should decrease while the fast edge should remain constant or increase.

The median profiles of slow and fast spectral edges separated for the three $DWR_{KaW}$ classes are shown in Figure 5. We can clearly see that the slow-down on the slow edge is larger than on the fast edge. This implies that the MDV slow-down is indeed
a combination of vertical air motion and the formation of a new, slower spectral mode. The reduction on the fast edge is almost





only visible for the largest $DWR_{KaW}$ class while the reduction on the slow edge is also noticeable for the lower $DWR_{KaW}$ classes. When focusing on the largest $DWR_{KaW}$ class and zooming into the temperature region of the DGL (Figure 5b), we can roughly estimate the velocity reduction at the two spectral edges. We assume that the spectral edges without new particle mode or updraft would remain constant or increase towards warmer temperatures. For each spectral edge profile, we search

for the level were we find the velocities indicating the onset of a slow-down (vertical red lines in Figure 5b). If we subtract the velocities at this level from the values of the strongest reduction found in the DGL, we obtain a lower limit estimate of the real slow-down. For the slow edge we find a reduction of 0.28 m s$^{-1}$ between –18 °C and –14 °C. On the fast edge, the reduction is almost a factor of three smaller with a reduction of 0.1 m s$^{-1}$ between –16 °C and –14 °C. The total slow-down in the MDV is 0.18 m s$^{-1}$ between –16 °C and –14 °C, which is less than for the slow edge simply due to the stronger contribution

of the larger (faster) particles to the MDV.

Even though the estimated upward air motion seen at the fast edge is smaller than the reduction of the slow-edge velocity, the presence of an updraft can also be seen at the slow edge. The median values reduce at –14 °C to almost 0 m s$^{-1}$; the quantiles indicated by the shaded areas in Figure 5b show even upward (positive) velocities which strongly indicates an upward air motion. The presence of an updraft was also found by Dias Neto (2021) for a similar winter dataset collected at JOYCE-CF.

Our estimated updraft velocity represents, however, only a lower boundary of the true updraft speed as the terminal velocity of the particles at the spectral edges are unknown.

### 4.3   Investigation of the small particle mode

The spectral analysis in the previous section clearly indicated the formation of a new, slow falling ice particle mode in the DGL. As polarimetric variables are known to be particularly sensitive to newly formed ice crystals with asymmetric shape, we

also sort the polarimetric profiles according to the $DWR_{KaW}$ classification in order to analyse their relation to the aggregation class.

Interestingly, the median ZDR profiles do not reveal a clear maximum of ZDR at –15 °C (Figure 6a) despite the expected presence of dendrites with very low aspect ratios. Between -30 and 0 °C we find the ZDR profiles to be increasingly shifted to smaller values for higher aggregate class. From the ZDR profiles alone it is very difficult to tell whether this shift is caused

by a change of small ice particle properties or by varying number, size, or density of aggregates. As already illustrated in the case study analysis (Figure 2h), the high-ZDR producing particles usually populate at lower Doppler velocities and are thus well separated from larger, low-ZDR producing aggregates. As a result, the $sZDR_{max}$ (Figure 6b) is mostly unaffected by the presence of aggregates and shows the ZDR signature of the most ZDR producing particles present in the radar volume. Increasing first at –18 °C, $sZDR_{max}$ of aggregate class 2 and 3 reaches a maximum within the bottom half of the DGL. Below

the DGL, $sZDR_{max}$ remains relatively constant between 1 and 1.5 dB down to –3 °C. Further down, $sZDR_{max}$ drops to values of 0.8 to 1 dB, which are similar to the values found at temperatures colder than –20 °C. Overall, the $sZDR_{max}$ in the DGL is only weakly dependent on aggregate class, with only slightly larger values (ca. 0.5 dB) for aggregate class 2 and 3 compared to class 1. This seems to indicate that the properties impacting ZDR, such as aspect ratio or ice density of the dendrites and other plate-like particles growing in the DGL, are overall relatively similar for aggregate classes 2 and 3. However, the position of





the maxima within the DGL of aggregate class 2 and 3 differ. Class 3 reaches its maximum slightly below $-15\,^\circ$ C, while class 2 reaches its maximum at the bottom of the DGL. This might indicate that the $sZDR_{max}$ producing particles of class 2 continue to grow and increase their aspect ratio and/or density throughout the entire DGL, while the particles in class 3 reach their largest aspect ratio already slightly below $-15\,^\circ$ C. Most notably, when comparing ZDR and $sZDR_{max}$, the $sZDR_{max}$ profiles lack the shift towards higher values with lower aggregate class. As both variables are independent of particle concentration, it

appears most likely that the ZDR shift is related to on average larger aggregates throughout the profile rather then less ZDR producing ice crystals. This interpretation is confirmed by the $DWR_{KaW}$ profiles in Figure 4b which show overall larger mean aggregate sizes between -30 and $0\,^\circ$ C for increasing aggregate classes. Apparently, the maximum size of aggregates in the DGL is correlated with the mean size of particles above the DGL.

    Similar to $sZDR_{max}$, the median KDP profiles (Figure 6c) above the DGL show only low values ($0.25\,^\circ$ km$^{-1}$) and no clear

difference between the aggregate classes. This implies that the particles falling from aloft into the DGL are not only similar in terms of their properties impacting ZDR, but also in terms of average concentration. All median KDP profiles start to increase at $-18\,^\circ$ C which is again very similar to the behaviour of $sZDR_{max}$. The KDP values within the DGL show a stronger increase for larger aggregate classes as observed for $sZDR_{max}$. Interestingly, the KDP profile observed for aggregate class 3 is on average not peaking directly at $-15\,^\circ$ C, but rather linearly increasing from $-18\,^\circ$ C towards $-12\,^\circ$ C where it reaches a maximum of

$0.7\,^\circ$ km$^{-1}$. The KDP profiles remain enhanced from the bottom of the DGL down to -3 $^\circ$ C where their values suddenly drop to nearly $0\,^\circ$ km$^{-1}$. The rapid decrease of KDP and $sZDR_{max}$ at temperatures larger then -3 $^\circ$ C is correlated with the increase of $DWR_{XKa}$ (Figure 4c) in this region. It appears to be quite likely that the second intensified aggregation zone close to $0\,^\circ$ C is mainly responsible for the reduction of high-ZDR producing ice particles.

    In order to complement the picture of the small particle evolution, in Figure 6d) we also included the skewness of the non-

polarimetric Ka-Band Doppler spectra recorded in zenith. Asymmetric broadening of the spectra on the fast edge (negative skewness) or on the slow edge (positive skewness) can be very well detected in the skewness profile (see also example profile in Figure 3). The formation of new, small ice particles can be expected to result in positive skewness values even if the spectra do not reveal a separated spectral mode. Above the DGL, all median skewness profiles are close to 0 indicating on average symmetrical spectra. Similar to KDP, the skewness values increase most rapidly at $-18\,^\circ$ C. We also find clearly larger skewness

values (up to 0.2) for higher aggregate classes. For the largest aggregate class, the skewness values also remain close to 0.2 down to the $-5\,^\circ$ C level. This similarity to the signature found in KDP is even more surprising as we can expect processes, such as riming (broadening on fast spectral edge) to decrease skewness, and hence potentially masking the signature of the small particle mode. Also, when looking at the 75% percentiles (shaded areas), we find, similar to $sZDR_{max}$ and KDP, distinct maxima at $-15\,^\circ$ C and $-5\,^\circ$ C.

Summarizing the results of this section, we can say that our statistics reveal that the aggregation in the DGL is correlated to growth and evolution of asymmetric particles within the DGL. Aggregation in the DGL appears to be stronger if larger aggregates fall already from higher levels into the DGL. Most notably, signatures related to crystal growth or aggregation that evolved in the DGL appear to persist to lower layers until the $-5\,^\circ$ C is reached. In the following section, we will discuss our main findings together with laboratory results and attempt a physical interpretation of the potentially involved processes.



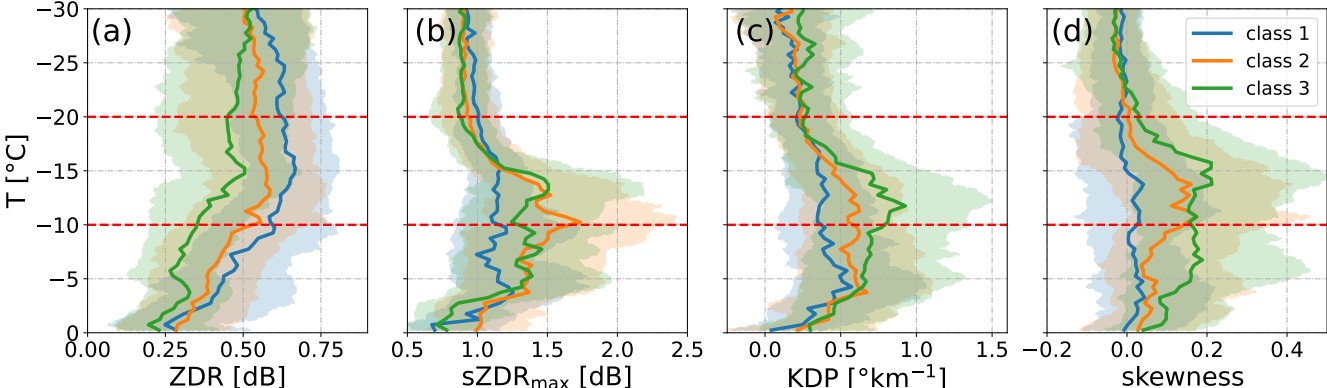

**Figure 6.** Same as in Figure 4, but showing profiles of W-Band (a) ZDR, (b) sZDR$_{max}$, (c) KDP, and (d) Ka-Band skewness (for interpretation of the skewness sign see text). Note that the polarimetric variables have been obtained at 30° elevation and the skewness from the zenith Doppler spectra.

## 5  Discussion

### 5.1  Interpretation of the temperature dependence of ice particles, aggregates, and vertical air motion in the DGL

The most striking features visible in the radar data of the DGL are rapid changes of vertical gradients within a relatively small temperature range, as well as distinct maxima at specific temperatures. In order to simplify their interpretation, we schematically show the radar profiles for the highest aggregation class in Figure 7, together with related results from previous laboratory studies.

According to (Takahashi, 2014), hereafter TKH14, the increase in mass and size of plate-like particles is strongest between –16 and –13 ° C with a local maximum at –15 ° C for stellar particles. When growing at constant temperature and high supersaturation, they develop distinct habits ranging from sectors (SEC), broad branched particles (BB), stellars (STEL), dendrites (DEN), and fern-like dendrites (FERN) (indicated on the left in Figure 7). This shape dependency on temperature was also confirmed by airborne in-situ observations (Bailey and Hallett, 2009). The plate-like particle growth connected to increasing sZDR$_{max}$ values seems to start at –20 ° C. This temperature roughly coincides with the temperature level where the aspect ratio of crystals observed in the laboratory starts to deviate from unity (see Fig. 2 in Takahashi et al., 1991, hereafter TKH91). Initially, the increase in sZDR$_{max}$ is relatively moderate down to –15 ° C. At this temperature level, sZDR$_{max}$ strongly increases reaching its maximum values between –13 and –14 ° C. As long as the particles grow in the plate-like growth regime, we can expect a certain correlation of aspect ratio and size. Regions showing enhanced sZDR$_{max}$ values are also likely connected to larger sized crystals. Interestingly, the maximum in sZDR$_{max}$ is found at slightly warmer temperatures then –15 ° C where the maximum size and aspect ratio is found in TKH14 and TKH91. A simple explanation might be the fact that unlike the particles grown in the experiments by TKH14 and TKH91, particles in real clouds are sedimenting into different temperature regions while growing. Once the sedimenting particles reach the –15 ° C level, they grow most efficiently by vapour deposition

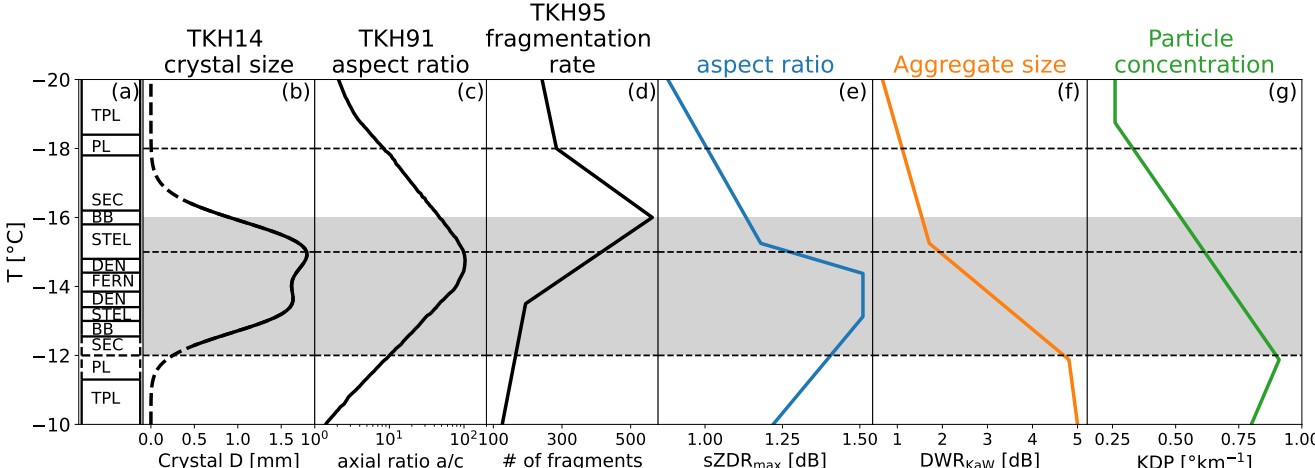

**Figure 7.** Schematic plot combining the main features found in the radar profile statistics (colored lines) discussed in Section 4 and results from laboratory experiments (black lines) in the DGL. The leftmost column denotes temperature regions where specific crystal types are growing (according to TKH91 and TKH14). The abbreviations used in this study and the ones according to the classification of Kikuchi et al. (2013) used in TKH14 denote: thick hexagonal plates (TPL, P1b), hexagonal plates (PL, P1a), sectors (SEC, P2a), broad branched (BB, P2b), stellars (STEL, P3a), dendrites (DEN, P3b), and fern-like dendrites (FERN, P3c). (a) The maximum crystal size and (b) aspect ratio (defined as ratio of the maximum dimensions along the a- and c-axis) which is reached in the experiments by TKH14 (their Table 1) and TKH91 (their Fig. 2) at constant temperature and liquid water saturation after 10 minutes growth time. The solid line indicates the temperature region in which the experiment was conducted, the dashed line shows the interpolation to warmer/colder temperatures which the fitted function to the measurement predicts. (c) Number of fragments collected after the collision of two ice spheres at different temperatures (fit to values shown in Fig. 14 in TKH95). Median profiles of observed (d) $sZDR_{max}$ (blue), (e) $DWR_{KaW}$ (orange), and (f) KDP (green) for the largest aggregate class shown in Figures 4 and 6. Note that the original profiles have been reduced to the main features such as maximum/minimum or strongest vertical change in order to simplify the discussion. The grey shaded area denotes the temperature region where the spectral edge analysis (Section 4.2) indicated upward air motion possibly related to latent heat release. The titles in (d)-(f) indicate the most common particle properties to which the polarimetric variables are related to (a more detailed discussion is provided in the text).

as indicated by the strong gradient found in the $sZDR_{max}$ profile. The most favourable growth region (indicated in the curve of maximum particle size found in TKH14) extends from –15 ° C to almost –13 ° C which is the location where $sZDR_{max}$ indicates the largest particles with most extreme aspect ratios. But why are $sZDR_{max}$ values not increasing further towards the bottom of the DGL? Scattering calculations (Myagkov et al., 2016; Hogan et al., 2000) indicate, that ZDR is only slowly increasing once a certain aspect ratio is reached (see also Figure C1b). The $sZDR_{max}$ values can therefore be expected to reach

a certain saturation value, once the particle grew into a shape with very low aspect ratio. With increasing size, also the particles' cross sectional area strongly increases which makes it more likely for the particle to collide with another crystal or aggregate. At a certain point, increasing aggregation (more likely for larger crystals) might counteract the general increase in crystal size leading to the observed slight decrease in $sZDR_{max}$ towards –10 ° C. The increasing $DWR_{KaW}$ values in the DGL appear to be





consistent with this interpretation. The sZDR$_{max}$ and DWR$_{KaW}$ most rapidly increase both at –15 ° C but unlike sZDR$_{max}$, the
DWR$_{KaW}$ continues to rise throughout the DGL. This effect might also be responsible for the different vertical location of the
maximum in sZDR$_{max}$ for aggregate class 2 and 3 (Figure 6b)). More intense aggregation (class 3) might consume the largest
dendrites earlier, which results in a sZDR$_{max}$ maximum closer to the –15 ° C level.

The temperature region of maximum crystal growth found in TKH14 also roughly coincides with the temperature region
where the upward air motion is found to be largest. This indicates that the weak upward air motion might indeed be a result
of the latent heat released by the intensified depositional growth. If we look again at the spectral edge velocities in Figure 5b,
we see that the slow edge velocity begins to decrease already at a temperature of –18 ° C which is almost 2 ° C colder then
the temperature where the fast edge is affected by the updraft. At –18 ° C the number of new particles appears to increase,
which can be also seen in the KDP profile. The latent heat release due to the increasing number and more favourable growth
conditions starting at –16 ° C might finally be sufficient to also increase buoyancy enough to sustain upward air motion. The
updraft might even cause a positive feedback, as its presence enables the particles to grow in the favourable growth region
longer. This would further enhance their mass uptake by deposition and increase the latent heat release.

The reverse explanation that, an updraft produces local enhancement of supersaturation leading to subsequent nucleation
and depositional growth, can not be entirely ruled out by the observations. However, it appears rather unlikely that a synoptic
or small-scale dynamical feature would be statistically prevalent in this narrow temperature region, despite the large number
and variety of cloud and weather conditions included in our statistics. We speculate that one reason why such a latent heat
driven upwind has not been detected so far in numerical weather prediction models might be simply related to the fact that
most models do not include an explicit habit prediction scheme which is probably needed to reproduce the intensive growth
rate found in the laboratory.

The profile of KDP, with its sudden increase at –18 ° C and its nearly linear increase towards –12 ° C, appears to be more
challenging to be explained by updraft and depositional growth features. If depositional growth alone was responsible for the
increase in KDP, we would expect the profile to have a similar shape as sZDR$_{max}$. KDP is well known to be closely related to the
concentration of asymmetric particles (Kumjian, 2013). Primary ice nucleation appears to be rather unlikely as an explanation
for increasing KDP in the DGL. The activation of INPs is expected to decrease with warmer temperatures (Kanji et al., 2017)
which is opposite to the KDP signature found in our dataset. The increase of KDP towards the bottom of the DGL is even more
surprising as ongoing aggregation (increasing DWR$_{KaW}$ values) should reduce the number concentration of KDP producing
particles, therefore reducing KDP or at the least keeping KDP constant.

Single scattering computations for realistically shaped crystals and aggregates show that KDP is by approximately a factor
of 3 larger for dendrites then for same-sized aggregate (Appendix C). Therefore, the contribution of aggregates to total KDP
can not be completely neglected. A simple calculation example for the bottom of the DGL using DWR, Ze and KDP values
for class 3 revealed that aggregates might contribute 1/3 to the total KDP (Appendix C). KDP at W-Band can be interpreted
in a similar way as commonly done for lower frequency radars. Unlike for Ze and ZDR, KDP seems to be not affected by
non-Rayleigh scattering effects.





Secondary ice processes (SIP, Korolev and Leisner, 2020; Field et al., 2017) appear to be a more likely explanation for the observed increase in particle concentration. Several in-situ studies (e.g., Rangno and Hobbs, 2001; Hobbs and Rangno, 1998, 1990, 1985) have reported fragments of stellars and dendrites in the DGL. The most likely SIP explaining this effect appears to be fragmentation of ice particles when colliding with each other. Unfortunately, this process has so far only been studied in three laboratory studies (Takahashi et al., 1995; Griggs and Choularton, 1986; Vardiman, 1978). The number of fragments ejected when colliding two ice spheres at constant temperature and water saturation has been found by Takahashi et al. (1995), hereafter TKH95, to be particularly enhanced inside the DGL between −18 and −12 ° C. Although the collisional energy from two cm-sized ice spheres used in TKH95 is quite unrealistic for the scenarios observed in our cases, the rapid growth of fragile dendritic arms on the ice spheres observed by TKH95 indicates that such structures are likely to also grow on aggregates which we found to be sedimenting from above into the DGL. It appears quite probable that collision or even touching of two aggregates might already cause fragmentation of the delicate structures growing on their surfaces. Griggs and Choularton (1986) also report that dendrite crystals can fragment without collisions simply due to air drag. Recent modelling studies suggest that ice collisional fragmentation can elevate the ice number concentration by three orders of magnitude (Georgakaki et al., 2022).

Clearly, more insights into the competing effects of aggregation and potential particle generation by ice fragmentation needs to be explored in model simulations which should include habit dependent growth and also recent formulation of ice fragmentation (Phillips et al., 2018).

## 5.2   Dependency of DWR and polarimetric quantities on cloud top temperature

Previous studies (Trömel et al., 2019; Griffin et al., 2018; Oue et al., 2018) found evidence that the cloud top temperature (CTT) is correlated with polarimetric radar features within the DGL. Griffin et al. (2018) analysed five winter storms at S-Band and found an increase of the 80th percentile of KDP values inside the DGL with colder CTT. Similar dependencies were also found for a large set of mid-latitude clouds (Trömel et al., 2019) and case studies of Arctic mixed-phase clouds (Oue et al., 2018). Griffin et al. (2018) explained the high KDP in the DGL with a high number of irregular crystals or nearly isometric aggregates falling from cloud top. At colder CTT the primary ice production can be expected to be larger due to the temperature dependence of INP activation (Kanji et al., 2017). At $T<-37$ ° C also homogeneous ice nucleation can further increase the number of ice particles. Although we found no clear correlation of aggregation in the DGL with particle concentration or habit of particles sedimenting from aloft in Section 4.3, we sorted our profiles with respect to CTT in order to allow a direct comparison with previous studies.

We determined cloud top height using the Ka-Band radar because it is the most sensitive radar and it is less affected by signal attenuation compared to the W-Band. The temperature information is taken again from the model analysis implemented in Cloudnet. All profiles with continuous values from cloud top down to −10 ° C level (to avoid multi-layer cloud cases) were sorted into three 10 ° C wide CTT regimes ranging from −60 to −30 ° C (Figure 8).

When focusing first on the region above the DGL (−30 to −20 ° C), the DWR$_{KaW}$ profiles for the different CTT (Figure 8b) show rather similar values around 0 dB. Also KDP values for different CTT are relatively similar above the DGL, varying



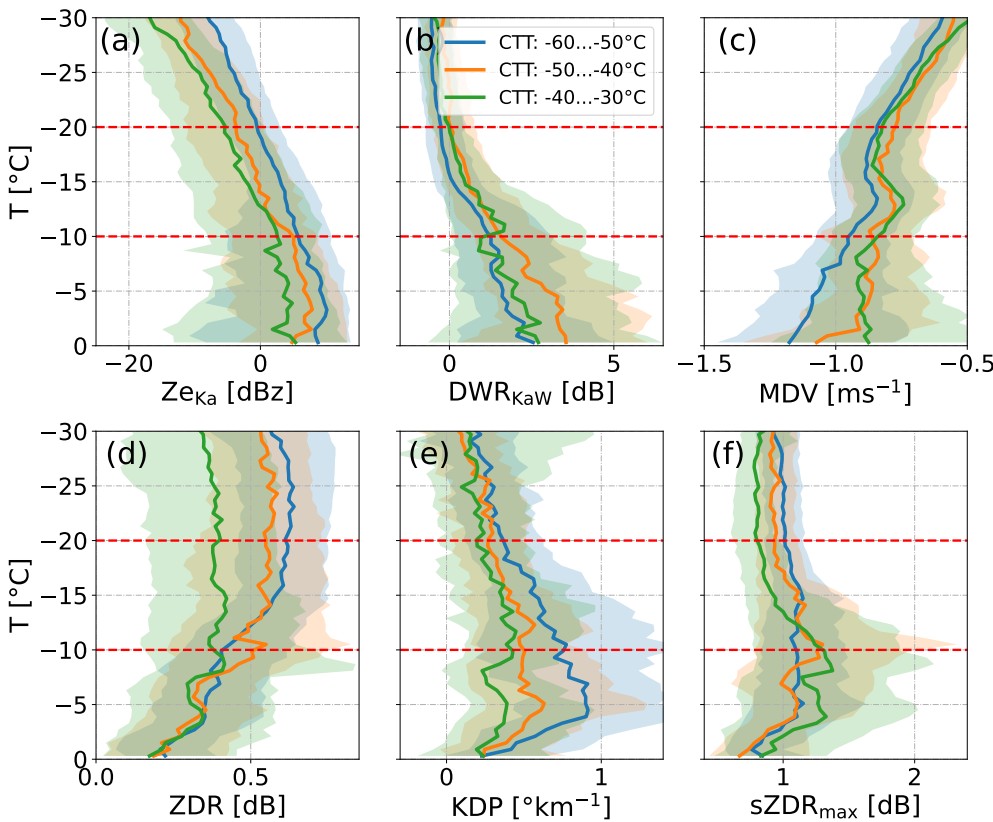

**Figure 8.** Dependency of (a) Ze , (b) $DWR_{KaW}$, (c) MDV, (d) ZDR, (e) KDP and (f) $sZDR_{max}$ on cloud top temperature (CTT). The profiles of each radar observable where stratified with temperature and classified into CTT classes. As in Figure 4, the solid lines depict the median of the distribution in each CTT-class. The shaded area indicates the quantiles of the distribution.

between 0.1 and 0.4 $°\,km^{-1}$ and showing maximum differences between the CTT regimes of less than 0.2 $°\,km^{-1}$ (Figure 8e). A slightly larger separation with CTT regime can be found for $sZDR_{max}$ and ZDR (Figure 8f,d). Colder CTT seem to be connected to slightly larger ZDR values. The maximum differences in ZDR and $sZDR_{max}$ for the three CTT regimes are both around 0.25 dB. Also Ze at Ka-Band (Figure 8a) shows larger values for colder CTT.

600 

From these observations we can conclude that the concentration of particles (KDP) and average size of aggregates or polycrystals ($DWR_{KaW}$) above the DGL does not substantially vary for different CTT. The similarity of the CTT dependency in ZDR and $sZDR_{max}$ rather indicates that the three CTT regimes lead to different shapes of ZDR producing crystals. This might simply be related to different habits that the particles grow into after being nucleated at different CTT. Particles falling from

605 lower CTT have also had more time to grow by deposition and can hence reach larger sizes and potentially more asymmetric shapes.





Inside the DGL, we find a continuous increase in KDP and rising values of sZDR$_{max}$ and DWR$_{KaW}$ especially at temperatures warmer than $-15\,^\circ$ C. Considering that the analysis was performed for the same cases as analysed in Section 4, the median profiles appear to be much less dependent on CTT as compared to sorting them with respect to the maximum DWR$_{KaW}$ in the DGL. Except for a clear slow-down in MDV, the signatures inside the DGL, for example the steep increase in sZDR$_{max}$ at $-15\,^\circ$ C, are much less pronounced. Still, certain weak dependencies on CTT can be found in some variables shown in Figure 8. The total ZDR decreases in the DGL with a larger reduction found for profiles starting at colder CTT. The KDP values increase stronger for colder CTT regimes. For CTT between $-60$ and $-50\,^\circ$ C, KDP increases from around $0.4\,^\circ$ km$^{-1}$ at $-20\,^\circ$ C to $0.7\,^\circ$ km$^{-1}$ at $-10\,^\circ$ C. The MDV slow-down is less pronounced for colder CTT. Below $-15\,^\circ$ C, the particles fall faster for colder CTT. Interestingly, the DWR$_{KaW}$ profiles show no evidence of a strong dependence of aggregation in the DGL on CTT regime.

The results we obtained by sorting the profiles according to their CTT reveal, similar to Section 4, that the main changes of particle concentration, crystal shape and aggregate size take place within the DGL. The role of particles sedimenting from upper layers appear to be small regardless of whether we sort by CTT or maximum DWR$_{KaW}$. This is in agreement with Dias Neto (2021), who found no clear dependency of aggregation strength on CTT. However, why do we still find larger KDP values for colder CTT if the influence of particles sedimenting from above is weak? Deeper clouds might simply provide overall a more favourable environment (for example larger updrafts, larger super-saturation) for ice particle growth. This might also lead to higher super-saturation inside the DGL. As a result, we expect primary ice nucleation as well as secondary ice processes to be more enhanced in such an environment. A more intense depositional growth of delicate dendritic structures will most likely also impact number and size of fragments caused by ice collisional fragmentation. As discussed in the previous section, there is growing evidence that fragmentation is a potential source for enhancing particle number concentration in the DGL. The dependence of KDP on CTT found in our study as well as in previous work might therefore be less related to the larger nucleation rates expected for colder CTT. The sorting by CTT might simply result into a separation of cloud regimes with more or less favourable growth conditions. This aspect should be further investigated with future campaign datasets that include a large number of reliable humidity profiles (e.g., from frequent radio soundings).

## 6 Summary and Conclusions

A statistical analysis of three months of ground-based, triple-frequency (X-, Ka-, W-Band) Doppler radar observations combined with polarimetric Doppler W-Band observations was conducted at the JOYCE-CF site in order to better understand growth signatures and related processes in the DGL. Similar to previous studies, we find rapid aggregation taking place in the DGL in combination with the formation of a new ice particle mode, most likely associated to dendritic particles.

After classifying the profiles with respect to their maximum average particle size (maximum DWR$_{KaW}$), we found a substantial reduction of the MDV in the DGL, which is strongest for the highest aggregate class. An analysis of the spectral edge velocities revealed that part of the reduction is due to a new mode of slow falling ice particles, that first appears in the spectra at $-18\,^\circ$ C. In addition, an updraft in the order of $0.1$ m s$^{-1}$ is revealed by the fast spectral edge velocity with a maximum





reached at –14 ° C. As suggested by previous studies, it appears most likely that the updraft is a result of latent heat release due to enhanced depositional growth in the DGL.

Clearly, processes in the DGL are strongly tied to temperature. After combining the main signatures revealed by spectral multi-frequency and polarimetric observations with recent laboratory experiments, we derived the following interpretation of particle evolution within the DGL:

– The concentration of particles indicated by KDP continuously increases from –18 to –12 ° C. This increase in concentration seems not to be strongly affected by the particle concentration falling from above. Only for the 75% percentile we find a distinct maximum of KDP at –12 ° C, similar to previous studies using lower frequency radars. The temperature where KDP first increases roughly coincides with the temperature where laboratory studies found an increase in the numbers of fragments ejected due to particle collisions. Such a SIP could potentially compensate the loss of particles 650 due to aggregation within the DGL and hence explain the continuous increase of KDP.

– The maximum spectral ZDR ($sZDR_{max}$) indicates that the aspect ratio of dendritic particles strongly increase at –15 ° C, coinciding with the temperature of maximum growth rate and aspect ratio measured in the laboratory. Slight temperature shifts between the radar observations and laboratory results can be most likely assigned to particle sedimentation while growing. Similar as for KDP, no strong difference is found in $sZDR_{max}$ for particles sedimenting into the DGL from 655 above. However, $DWR_{KaW}$ and ZDR indicate that aggregates sedimenting from higher altitudes into the DGL are larger for cases with enhanced aggregation in the DGL.

– The temperature region where the analysis of the spectral edge velocity indicated an updraft coincides with the region of strongest depositional growth and increase in mean aggregate size. This updraft, potentially connected to latent heat release, might cause a positive feedback as it would enhance the residence time of small particles in the favourable 660 growth zone.

Sorting the profiles with respect to cloud top temperature revealed only slight differences in ice particle shape but nearly negligible differences in concentration or mean size for particles entering the DGL from above. The strongest change in concentration, aspect ratio and mean aggregate size is again observed within the DGL. This highlights the importance of processes taking place inside the DGL for evolution of particle concentration and size. Larger aspect ratios and sizes of ice 665 particles falling from above into the DGL as well as the generally stronger increase of KDP in the DGL for colder CTT might be simply explained by the overall more favourable growth conditions expected for deeper cloud systems, such as higher super-saturation.

Our statistical analysis further revealed that KDP and $sZDR_{max}$ values remain enhanced after leaving the DGL down to -3 ° C where their values rapidly decrease towards the melting layer. Other SIP being active at warmer temperatures might be a 670 potential source for new ice particles that maintain the high KDP values. Intensified aggregation at temperatures warmer than -3 ° C is the most likely explanation for the rapid decrease of polarimetric variables with concurrent increase of DWR. This increase in aggregation might be explained by the strongly increasing thickness of a QLL on ice surfaces, which is expected to increase the sticking efficiency of all ice particles.





This study clearly demonstrates the added value of combining different radar approaches including Doppler spectral analysis, high-frequency radar polarimetry, as well as multi-frequency observations for ice microphysical studies. A statistical analysis as presented in this work can provide robust estimates of potential correlations between different radar variables and their specific temperature dependency. Unlike case studies, such a statistical approach also provides an estimation of the natural variability of observables involved. Statistically based observational process signatures are very useful for evaluating and improving microphysical schemes in weather prediction models (e.g., Karrer et al., 2021; Ori et al., 2020). They are also urgently needed as constraint for recent model developments such as habit dependent growth (Jensen et al., 2017; Sulia and Kumjian, 2017; Harrington et al., 2013; Hashino and Tripoli, 2007) and Langrangian Monte-Carlo models where the particle history can be traced (Grabowski et al., 2019; Brdar and Seifert, 2018). Signatures in remote sensing datasets can also guide new laboratory studies which are inevitably needed to improve our process understanding as well as microphysical parametrisations in models.

*Code and data availability.* The quality processed level 2 dataset is available on zenodo under the DOI: 10.5281/zenodo.5025636. The dataset that was used for this statistical analysis (5 minute temporal average containing all polarimetric variables as well as the zenith variables and variables derived from the Doppler spectra) is available under 10.5281/zenodo.5025636 . Due to the large size, the full level-0 dataset, containing the Doppler spectra of the four radars, is only available on request. The code used to reprocess, classify and plot is available in https://github.com/OPTIMICe-team/DGL-analysis-ACP-2022

### Appendix A: Chirp tables of the vertically pointing W-Band and slant polarimetric W-Band radar.

The range resolution, number of spectral averages, Doppler velocity resolution and Nyquvist range vary with range for the two FMCW W-Band radars. This is due to different chirp settings being defined for different range gate regions. The details of the chirp settings applied during TRIPEx-pol are summarized in Tables A1 and A2.

**Table A1.** Chirp table for the vertical W-Band radar installed during the TRIPEx-pol campaign.

| Specifications | Chirp 1 | Chirp 2 | Chirp 3 | Chirp 4 |
|---|---|---|---|---|
| Range [m] | $215-1475$ | $1482-3986$ | $3999-8151$ | $8165-11998$ |
| Range Resolution [m] | 36.0 | 12.8 | 12.8 | 12.8 |
| Number of Spectral Averages | 13 | 13 | 15 | 11 |
| Doppler velocity resolution [m s$^{-1}$] | 0.04 | 0.027 | 0.028 | 0.029 |
| Nyquvist range [m s$^{-1}$] | $\pm 10.26$ | $\pm 6.85$ | $\pm 3.41$ | $\pm 1.81$ |





**Table A2.** Chirp table for the polarimetric W-Band radar installed during the TRIPEx-pol campaign

| Specifications | Chirp 1 | Chirp 2 | Chirp 3 |
|---|---|---|---|
| Range [m] | $107-715$ | $751-5902$ | $5938-17994$ |
| Range Resolution [m] | 35.8 | 35.8 | 35.8 |
| Number of Spectral Averages | 28 | 56 | 104 |
| Doppler velocity resolution [m s$^{-1}$] | 0.05 | 0.042 | 0.039 |
| Nyquvist range [m s$^{-1}$] | $\pm 6.35$ | $\pm 4.98$ | $\pm 2.66$ |

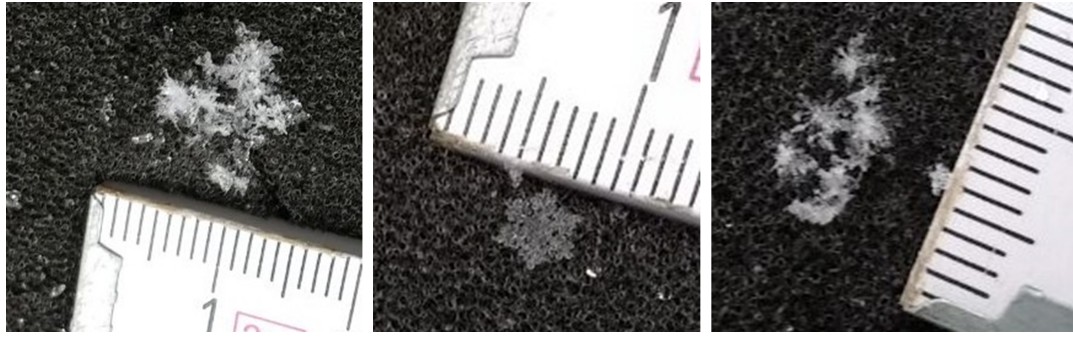

**Figure B1.** Example photographs of dendritic crystals and aggregates reaching the surface during the snowfall event occurring on 30th January 2019 at JOYCE. The sample pictures where taken at 09:00 UTC. The long ticks on the scale denote 1 cm while the short ticks denote 1 mm.

## Appendix B: Snow particle observations on the surface and zoomed view on Doppler spectra observed on 30th January 2019 at JOYCE-CF

On January 30th 2018, snowfall was observed on the ground at JOYCE-CF. Between 09:00 and 10:00 UTC, large dendritic crystals and aggregates could be photographed (Figure B1). The pictures are meant to complement the remote sensing observations from the case study presented in Section 3.

For a better visibility of the second spectral mode described in Section 3, we show a zoomed view of the spectral Ze in the DGL (Figure B2a)) and a single Doppler spectrum extracted close to the $-14\,^{\circ}$C level where the second, slow falling mode 700 can be recognized (Figure B2b)).

## Appendix C: Comparability of W-Band and X-Band polarimetric observations and estimation of the contribution of aggregates to KDP

The majority of polarimetric observations in the DGL have been obtained by lower frequency systems (e.g., S-, C-, X-Band). Only during recent years, an increasing number of higher frequency polarimetric cloud radars (mainly Ka- and W-Band) 705 became available. The use of higher frequencies has a number of advantages, such as larger KDP (increasing with $\lambda^{-1}$) for a





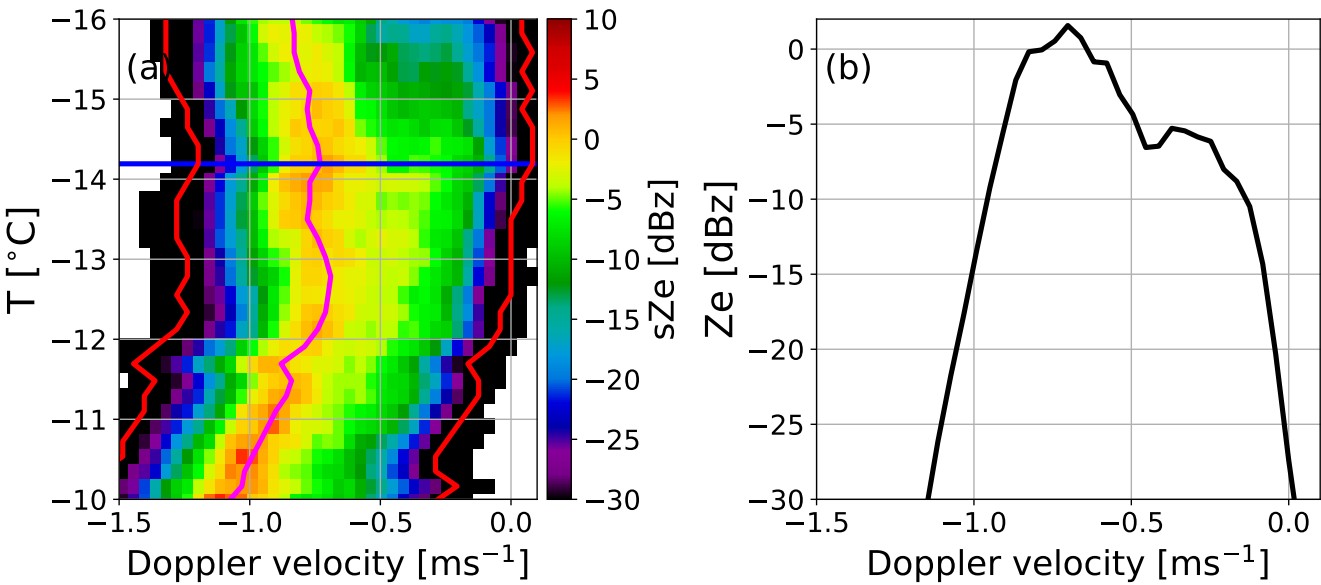

**Figure B2.** Zoom into the spectrogram shown in Figure 3: (a) spectral Ze from the Ka-Band radar, the magenta line depicts the position of the maximum of the main spectral mode. As in Figure 3d), the red lines indicate the derived spectral edge velocities. The blue line in (a) shows the temperature from which the single spectrum in (b) was taken.

given particle population. However, also non-Rayleigh scattering effects become increasingly important at higher frequencies which can make the interpretation of high-frequency polarimetric observations more challenging.

Certainly, an in-depth discussion of the differences in polarimetric observations at various frequencies is out of the scope of this study. However, a comparison of the size and frequency dependence of some key radar variables, such as Ze, ZDR, and
710 KDP, for single particles can help to understand which variables are more or less affected by non-Rayleigh scattering effects. Figure C1 shows those three variables which were derived using a recent scattering database by Lu et al. (2016) which also contains scattering properties needed for polarimetric quantities. As our focus in this study is on the DGL, we focused on horizontally aligned branched planar crystals and aggregates of stellars (HD-P1d) at an elevation angle of 30° (consistent with our observations during Tripex-pol). In addition to W-Band, values are calculated for X-Band as this is the lowest frequency
included in the database.

Up to 1 mm particle size the single-particle Ze at X- and W-Band are almost identical as expected from Rayleigh theory (Figure C1(a)). The slight differences between crystals and aggregates are most likely due to their different mass-size relations. No specific resonance effects are visible at X-Band up to 10 mm size while at W-Band we find the first distinct minimum at 3 mm which is close to the wavelength (3.3 mm). As a result, the Ze at W-Band is lower then at X-Band for particles larger
than 1 mm which is the reason for increasing DWR at larger mean size.

Similar resonance phenomena can also be found in ZDR at very similar particle sizes (Figure C1(b)). ZDR at X- and W-Band are very similar up to 1 mm size with larger values (3 dB) for crystals and smaller values (1.3 dB) for aggregates owing to their



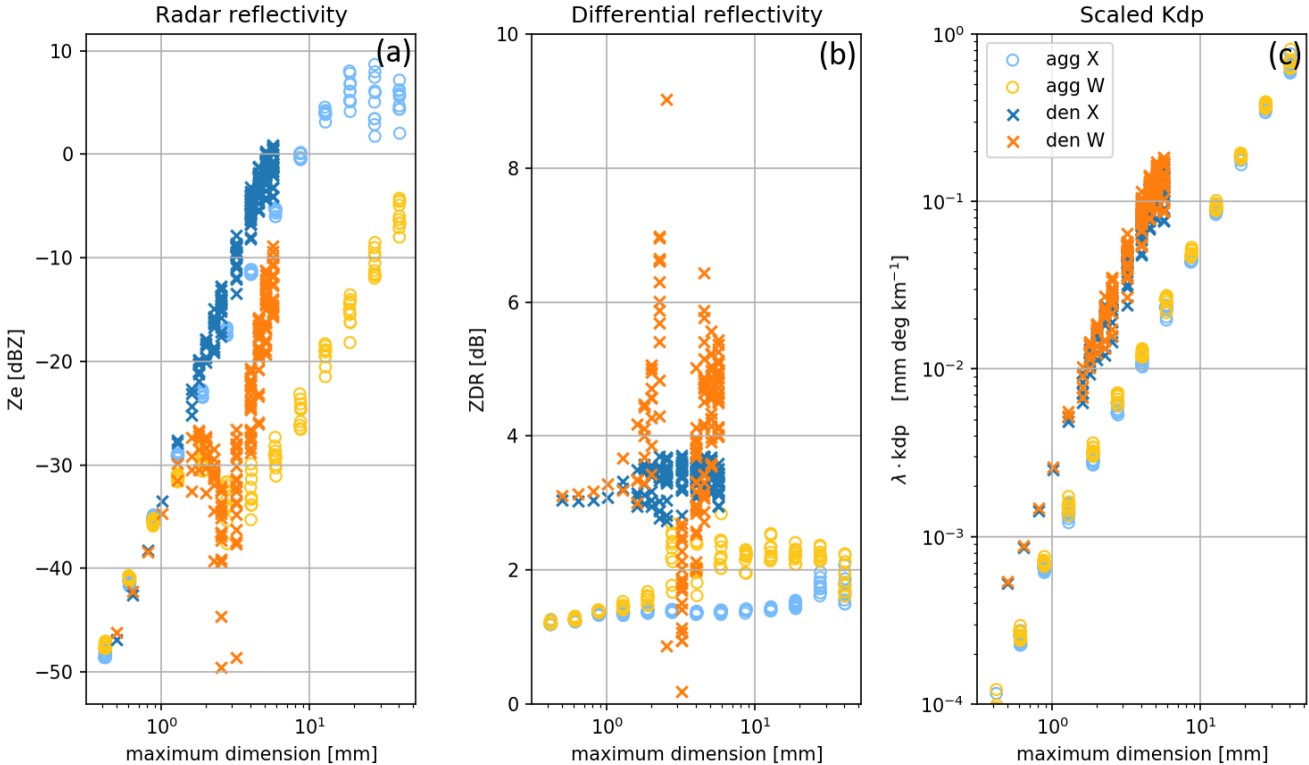

**Figure C1.** Scattering properties of single particles for X- and W-Band calculated with dendrites (branched planar crystals, blue and orange crosses) and aggregates of stellar crystals (HD-P1d, light blue and yellow circles) from the scattering database presented in Lu et al. (2016). The calculation of vertically pointing Ze for a single particle at X and W-Band is shown in (a), the ZDR at X- and W-Band and 30° elevation in (b) and the single particle KDP at X- and W-Band in (c). KDP was scaled with the wavelength $\lambda$ to allow a better comparison of the diameter dependent behaviour at X and W-Band.

lower density and less extreme aspect ratio. The ZDR values at X-Band remain relatively constant over the entire size range. At W-Band we find in addition to a strong minimum at 3 mm two distinct maxima at 2 mm and 4-5 mm. Similar but overall

less extreme resonance phenomena can also be found for the aggregates at W-Band. The ZDR at W-Band from the aggregates seems also to increase with particle size.

Despite the large values in ZDR, which can be reached at specific diameters due to resonance phenomena at W-Band (between 0 and 9 dB), the differences in total ZDR between X- and W-Band can be expected to be relatively small as the extreme values are likely to cancel out when integrating over a PSD (Matrosov, 2021). Even for spectrally resolved ZDR

resonance effects might be difficult to detect as various ice particle sizes are likely to fall into the same Doppler velocity bin due to natural variability in particle shape and orientation. Matrosov (2021) measured ZDR at W- and Ka-Band in Arctic clouds



and found that the ZDR differences are slightly increasing with ZDR but on average the differences are found to be less than 0.5 dB.

Most interesting for the interpretation of the results of our study is the comparison of single-particle KDP shown in Figure C1(c). One can see that after scaling KDP with $\lambda$, there are only very small differences found between X- and W-Band. Especially, no resonance phenomena as observed for Ze and ZDR can be found for KDP at any size and for both particle types. This is in agreement with Lu et al. (2015), who showed that simulated KDP at cloud radar wavelengths does not exhibit resonance phenomena. Also measured KDP values at Ka- and W-Band reveal only the expected wavelength scaling (Matrosov, 2021). Also the strong increase of KDP with particle size is remarkably similar at X- and W-Band (Figure C1(c)). As expected, the KDP for crystals is much larger (up to one order of magnitude at 4 mm) than for aggregates. However, as shown in the following simple calculation, the contribution of aggregates to the total KDP can usually not be neglected.

In the observational statistics (Figure 7) we saw that KDP and DWR continuously increase towards the bottom of the DGL. Can those enhanced KDP values maybe entirely be explained by the contribution from aggregates as for example suggested by Moisseev et al. (2015)? We try to shed light on this question with the following simple calculation.

For simplicity, we assume an inverse exponential size distribution for the aggregates of the form

$$N(D) = N_0 \cdot exp(-\Lambda D) \tag{C1}$$

with the slope parameter $\Lambda$ in $m^{-1}$, the intercept parameter $N_0$ in $m^{-4}$, and the particle size $D$. It appears reasonable to assume that at the bottom of the DGL $DWR_{KaW}$ and Ze are dominated by the contributions from the aggregates. Using the scattering properties shown in Figure C1, we can directly estimate $\Lambda$ to be $(2.25 \cdot 10^{-3})^{-1} m^{-1}$ for the maximum $DWR_{KaW}$ of $4.3 dB$ observed at –12 ° C for the largest aggregate class. With this $\Lambda$ we need to assume $N_0$ to be $5.6 \cdot 10^4 m^{-4}$ in order to match the Ka-Band Ze of 10.2 dBz at –12 ° C.

The W-Band KDP caused by this aggregate distribution is 0.28 ° $km^{-1}$ which is roughly one third of the observed KDP. If we repeat the same calculation with the 75th percentile of the radar variables measured at –12 ° C, we obtain for a $DWR_{KaW}$ of 6 dB, and Ze of 13 dBz, a KDP of 1.5° $km^{-1}$. The aggregates contribute still 0.32 ° $km^{-1}$ (20%) to the total KDP.

Unfortunately, we have less constraints on the PSD for the small, presumably dendritic particles at the –12 ° C level. For the case study shown in Figure 2 we observed on the ground dendrites reaching up to 5 mm size (see example in Figure B1). Cloud chamber experiments by TH14 show that particle sizes of 1-1.5 mm are reached in the DGL temperature regime after a growth time of 10 minutes. In order to produce the remaining KDP signal for the median KDP value at –12 ° C, a concentration of 2500 dendrites per $m^3$ with 1 mm or 120 per $m^3$ with a size of 5 mm would be needed. For the 75th percentile of the KDP observed at –12 ° C the concentration would increase to 4150 and 200 per $m^3$, respectively. For comparison, the expected number of ice nucleating particles at –12 ° C ranges between 1000 and 2000 $m^{-3}$ (e.g. DeMott et al. (2010)).

*Author contributions.* Data analysis, post-processing, generation of figures was performed by LvT with contributions from JDN and AM. LvT and SK conceptualized the methods and interpretation. LvT and DO performed the scattering calculations. LvT and SK prepared the paper with contributions from all co-authors.



*Competing interests.* The authors declare that they have no conflict of interest.

*Acknowledgements.* Contributions by J. Dias Neto, D. Ori, and S. Kneifel were funded by the Deutsche Forschungsgemeinschaft (DFG, German Research Foundation) under grant KN 1112/2-1 and KN 1112/2-2 as part of the Emmy-Noether Group "Optimal combination of Polarimetric and Triple Frequency radar techniques for Improving Microphysical process understanding of cold clouds" (OPTIMIce). The TRIPEx-pol campaign and work provided by L. von Terzi have been supported by the DFG Priority Program SPP2115 "Fusion of Radar

Polarimetry and Numerical Atmospheric Modelling Towards an Improved Understanding of Cloud and Precipitation Processes" (PROM) under grant PROM-IMPRINT (project number 408011764). We thank the Regional Computing Center of the University of Cologne (RRZK) for providing computing time on the DFG-funded (Funding number: INST 216/512/1FUGG) High Performance Computing (HPC) system CHEOPS as well as support. L. von Terzi and J. Dias Neto also acknowledge support from the Graduate School of Geosciences of the University of Cologne. The authors are indebted to staff of the University of Cologne, research center Jülich, and RPG, especially B. Bohn,

R. Haseneder-Lind, P. Krobot, B. Pospichal and A. Saljihi for their help with the installation of the W-Band radars, and K. Schmidt for the preparation of the scanning polarimetric W-Band radar for the TRIPEx-pol campaign.



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
