# Peer review of "Ice microphysical processes in the dendritic growth layer: A statistical analysis combining multi-frequency and polarimetric Doppler cloud radar observations"

_Atmospheric Chemistry and Physics, 2022_

## Referee Comment (RC1)

**Review of "Ice microphysical processes in the dendritic growth layer: A statistical analysis combining multi-frequency and polarimeteric Doppler cloud radar observations"**
**by von Terzi et al.**

The study by von Terzi et al. examines various microphysical processes that may contribute to the radar reflectivity, dual-wavelength ratio, mean doppler velocity (MDV) and polarimetric (ZDR and KDP) signals within the dendritic growth layer between -20 and -10$℃$. This work expands upon the TRIPEx campaign observations with Doppler spectra and polarimetric measurements that are known to be influenced by the presence of habits such as dendrites, aggregates, and needles. The manuscript is exceptionally written, well-referenced, and detailed in the methodology used. My main criticism revolves around some of the claims of the microphysical processes "observed" without actual in situ microphysical measurements. Nonetheless, I believe that the manuscript should be accepted for publication in ACP once the following Major and Minor comments are addressed.

**Major Comments**

1.  Some areas of the text containing claims of the microphysical processes observed (e.g., aggregation, SIP) could be scaled back somewhat given the absence of in situ microphysical observations. Further, measurements at the ground (e.g., Fig. B1) should be regarded with caution as they are the result of complex particle growth histories subject to hydrometeor transport by the horizontal wind and vertical motions, and as such are difficult to associate with radar measurements valid several km above ground. Some of the minor comments below address areas where text should be reworded to acknowledge these limitations or investigated further to strengthen the claims made.
2.  Discussion around DWR/polarimetric variables and cloud top temperature (CTT) can probably be flushed out more. First, DWR and KDP would probably be more affected by the depth below cloud top (e.g., better linked to residence time) than the CTT. Second, using Ka-band echo top heights (prone to some attenuation especially with rain below a bright band) and temperature from Cloudnet to get the CTT seem like compounding error sources. Was there satellite data to deduce the CTT? If not, it could be good to quantify or estimate the uncertainty with these CTT values, or add depth below cloud top to the discussion. Third, while the Ka-band radar has a high sensitivity, are you able to comment on any discrepancies between echo top and cloud top height? In other studies where airborne lidar measurements exist, the cloud top and echo top height (e.g., Ka-band) has been known to differ by a couple to a few hundred meters (e.g., generating cells).

**Minor Comments**

1. L2: "likely also" → "perhaps"
2. L92: "provide also" → "provide"
3. L302–304: Hydrometeor transport (e.g., horizontal wind) should be noted here.
4. Fig. 2: Why were -30 and -15℃ the contour levels chosen? It seems like -20 and -10℃ would better fit the DGL narrative and be consistent with the subsequent profile figures. Can you also comment on the cause for the gap in DWR measurements around 0700 UTC (panel c)?
5. L325: Are you able to cite a previous study that corroborates your claim/postulation?
6. L335: I'm not sure parentheses are needed around sDWR. Maybe turn this phrase into a list?
7. L343: Seems more like -7 or -8℃.
8. Table 2 and relevant discussion: Perhaps you can acknowledge that the D0 estimates are probably underestimated, particularly for the largest DWR class, as studies such as Mason et al. (2019; https://doi.org/10.5194/amt-12-4993-2019) find PSD shape is an important factor in the triple-frequency radar signature (and by extension in the DWR measurements).
9. L380: "temperature," → "temperature"
10. Fig. 4b: Can you comment on the negative DWR values above the DGL? Can calibration uncertainty (i.e., Section 2.2.3) attribute to this?
11. L398: Specify "more negative" in addition to slightly larger for clarity.
12. L400–401: Add negative signs to your MDV values for consistency.
13. L405: I'm unsure what you mean by "upwind" as it relates to large scale lifting. Can you clarify?
14. L429: Is this difference between the slow-down on the slow and fast edges in Fig. 5a statistically significant, or at least greater than potential uncertainties in the MDV measurements?
15. L456–457: True, but is the case study representative of the entirety of the project as it relates to ZDR?
16. L471: "confirmed by" → "consistent with"
17. L472–473: Remove "apparently", "DGL is correlated" → "DGL appears to be correlated"
18. L483: The study of Moisseev et al. (2015) is briefly mentioned in Appendix C, but it could be good to relate their findings to what's discussed in this paragraph.
19. L498: I think a word was left out for "until the -5℃ is reached". Possibly rephrase?
20. L506: Parentheses should be like "Takahashi (2014)".
21. L510, 642, and possibly elsewhere: You should also acknowledge ice supersaturation as an important factor for the ice crystal habit, as is mentioned in their study.
22. L516: "then" → "than"
23. L535–543: Have you looked at or considered other (i.e., mesoscale) effects? You mentioned earlier that many of these events were frontal driven, so it's plausible that frontogenesis or weak/elevated instability can contribute to regions of vertical motion in addition to latent heating by depositional growth.
24. L548: I agree that most bulk schemes lack the ability to resolve these growth rates and latent heating properly. Maybe add that bin schemes (e.g., Lee & Baik 2018;

https://doi.org/10.3390/atmos9120475) produce greater latent heat via deposition that leads to stronger updrafts.

25. L560 and Appendix C: Have you examined the per-particle KDP for other habits in the scattering database? Since mid-latitude winter systems typically consist of many habits in a radar gate, the sensitivity of KDP by habit as shown in Fig. C1c may mean that the 1/3 contribution of aggregates to the KDP signal may have large uncertainties.

26. L587: DeMott et al. (2010; https://doi.org/10.1073/pnas.0910818107) also has comprehensive (global) INP measurements.

27. Conclusions: I also think you should acknowledge that future studies employing in situ microphysical measurements could be a unique opportunity to validate the findings or confirm the speculated processes presented in this study.

---

## Author Response (AR1)

We would like to thank the reviewers for their time and effort reviewing our study. We have found the comments to be constructive and helpful and think that they have helped to improve our study! The comments of the reviewers are marked in black, our answers to the reviewers in red and the new text and lines of the revised document where the adjusted text can be found in blue. In the revised document, all new text is marked in blue, and deleted text is crossed out in red.

**Answers to reviewer 1**

**Major comments:**

1. Some areas of the text containing claims of the microphysical processes observed (e.g., aggregation, SIP) could be scaled back somewhat given the absence of in situ microphysical observations.

A: We completely agree with the reviewer, that it would be ideal to have additional proof by in-situ observations collocated with the radar observations. However, we believe that our interpretation of radar signatures in terms of underlying microphysical processes are well-grounded on previous studies where for example radars and airborne in-situ observations have been matched for case studies. In addition, a number of laboratory studies provide detailed information on which processes and particle habits can be expected for example at certain temperatures. Practically, it would be extremely expensive to collect airborne in-situ data for the many cases that we include in our statistics. We would like to emphasise that we see the strength in our analysis in the statistical approach instead of interpreting single case studies. Nevertheless, we added the following text in section 5.1 acknowledging the lack of in-situ observations for additional proof of process interpretation.

New Text in Line 572-575: *Although we are lacking in-situ observations from inside the cloud for our dataset, we base our interpretation on well-established relations between microphysical processes and distinct radar signatures. For example, aggregation can be clearly associated with an increase in DWR (e.g. Ori et al., 2020; Barrett et al., 2019; Dias Neto et al., 2019; Mason et al., 2019), whereas plate-like particle growth is strongly linked to enhanced ZDR and KDP (e.g. Moisseev et al., 2015; Schrom et al., 2015)*

Further, measurements at the ground (e.g., Fig. B1) should be regarded with caution as they are the result of complex particle growth histories subject to hydrometeor transport by the horizontal wind and vertical motions, and as such are difficult to associate with radar measurements valid several km above ground. Some of the minor comments below address areas where text should be reworded to acknowledge these limitations or investigated further to strengthen the claims made.

A: We agree that the measurements on the ground as provided in Fig. B1 should not be regarded as a proxy for the microphysical processes happening in the cloud. We have adjusted the sentence accordingly to avoid an over-interpretation of the snowflake pictures provided.

New Text in Line 322-326: *Certainly, the particles observed on the surface are not necessarily representative for the particles sampled with the radars due to impacts of advection and further growth processes during sedimentation towards the surface. Still, the snowfall reaching the ground was mostly comprised of unrimed or only slightly rimed crystals and aggregates. Visual observation on the ground at the site between 9 and 10 UTC (Figure B1) reveal the presence of stellar and dendritic crystals reaching up to 4 mm in size mixed with unrimed aggregates with maximum sizes up to 10 mm.*

2. Discussion around DWR/polarimetric variables and cloud top temperature (CTT) can probably be flushed out more. First, DWR and KDP would probably be more affected by the depth below cloud top (e.g., better linked to residence time) than the CTT.

A: We agree that the residence time of the particles is an important factor especially for DWR as the growth time will probably most affect PSD and mean size. Still, we focussed on CTT for two main reasons: 1) Previous studies showed some evidence that the aggregate size in the DGL is correlated with CTT - a connection which we wanted to evaluate with our larger dataset. 2) There is indeed a strong temperature dependence of primary ice nucleation and temperature which should impact the initial ice particle concentration. We prefer to leave the suggested investigation on additional parameters for future studies because the current manuscript is in our opinion already quite long. In addition, such an analysis would also require to take effects such as turbulence and advection of particles into account, which is not a trivial task.

Second, using Ka-band echo top heights (prone to some attenuation especially with rain below a bright band) and temperature from Cloudnet to get the CTT seem like compounding error sources. Was there satellite data to deduce the CTT? If not, it could be good to quantify or estimate the uncertainty with these CTT values, or add depth below cloud top to the discussion.

In order to adress possible uncertainties in the temperature information, we performed a comparison of the cloudnet temperature product to 27 radiosondes that were launched during the campaign. In general with a correlation of 0.9, a bias of 0.2°C and standard deviation of the bias of 1.1°C, the temperature information from cloudnet showed good agreement with the measured temperature (see Figure 1 below). We have added a new subsection describing this comparison of cloudnet and radiosonde temperature (see new section 2.4). We expect the impact of random uncertainties in the temperature information to have a negligible effect on our statistical analysis. We also agree that cloud top height might be biased low due to attenuation effects. However, we are quite confident that our CTH estimates are

reasonable. Attenuation at Ka band is less severe then at W band and fortunately the rain rates during the campaign were moderate enough so that we don't expect large attenuation effects. This is also confirmed by comparing the DWRs between X and Ka at cloud top which would be enhanced in situation of strong Ka band attenuation.

[Figure]

Figure 1: comparison of cloudnet and radiosonde temperatures during the TRIPEx-pol campaign. (a) scatterplot between cloudnet and radiosonde temperatures, (b) radiosonde temperature against the difference in cloudnet and radiosonde temperature

Third, while the Ka-band radar has a high sensitivity, are you able to comment on any discrepancies between echo top and cloud top height? In other studies where airborne lidar measurements exist, the cloud top and echo top height (e.g., Ka-band) has been known to differ by a couple to a few hundred meters (e.g., generating cells).

Unfortunately, we do not have airborne lidar measurements. We agree that with our method we likely underestimate the cloud-top height (CTH). For the analysis of the CTT, we chose relatively wide CTT bins (10°C bin width). We therefore do not think that a potential underestimation of the CTH by a few hundred metres or the over/underestimation of the temperature at cloud top would change the results significantly. We have tested the sensitivity of the statistical analysis of the CTT classes by perturbing the estimated CTH by several hundred metres. The Figure 2 and 3 below show the identical statistical analysis when adding an offset of 100 or 500m to the estimated CTH. Even with such a big perturbation, the medians of the different CTT classes still show similar features as described in the manuscript. We therefore think that a potential underestimation of the CTH does not change the interpretation of the CTT classes as included in the original manuscript. We have also tested the impact of possible under-and overestimation of the temperature at cloud top. We have added a bias of upto +-2°C to the CTT. Similar to changes in

CTH, a change in CTT has only minor effects on our results of the statistical analysis (Figures are not shown here).

[Figure]

Figure 2: Cloud top temperature analysis as is provided in Figure 8 in the manuscript. However, we added a height offset of 100m to the estimated cloud top height, in order to visualize possible impact of an underestimation of the cloud top height.

[Figure]

Figure 3: Same as Figure 2 but for a height offset of 500m.

**Minor comments:**

1. L2: "likely also" → "perhaps"

A: we have changed likely also to potentially

2. L92: "provide also" → "provide"

A: Changed in the text

3. L302–304: Hydrometeor transport (e.g., horizontal wind) should be noted here.
A: We have added possible advection of hydrometeors to this paragraph. The new text is written below the answer to the major comment 1 and can be found in line 322-326 in the revised document.

4. Fig. 2: Why were -30 and -15°C the contour levels chosen? It seems like -20 and -10°C would better fit the DGL narrative and be consistent with the subsequent profile figures. Can you also comment on the cause for the gap in DWR measurements around 0700 UTC (panel c)?

A: We agree with the suggested contour levels and we changed the figure accordingly. However, we kept the -15°C contour level because it allows to see that below -15°C the DWR and sZDRmax are enhanced. The gap in the DWR measurements is due to a high variance flag obtained by the data processing routine which causes the zenith W-Band radar data to be excluded. This flag indicates that during the relative DWR calibration described in Section 2.2.3, the variance in time of the calculated DWR between Ka and W-band were larger than 2dB^2. (See also Dias Neto et al., 2019 https://essd.copernicus.org/articles/11/845/2019/)

5. L325: Are you able to cite a previous study that corroborates your claim/postulation?
A: There are in general only very few studies that looked into spectral DWR. Our reasoning here is simply based on the fact that the maximum sDWR (independent on concentration) is not coinciding with maximum sZe (driven mainly by size, density, and concentration).

6. L335: I'm not sure parentheses are needed around sDWR. Maybe turn this phrase into a list?
A: Changed in the manuscript
New Text in line 355 in revised document: *For temperatures warmer than -15°C, the fall velocities, sZe and sDWR-KaW of the secondary mode increase…*

7. L343: Seems more like -7 or -8°C.
A: We agree that the weak secondary mode in sZe is closer to -7 or -8°C, however the increase of sZDR seems to start closer to -10°C. We adjusted the text accordingly.
New Text line 365-367: *At temperatures around -10°C, the sZDRmax values increase again. The maximum in sZDRmax at around -8°C roughly coincides with the appearance of a weak secondary mode in sZDR and an increase in KDP*

8. Table 2 and relevant discussion: Perhaps you can acknowledge that the D0 estimates are probably underestimated, particularly for the largest DWR class, as studies such as Mason et al. (2019; https://doi.org/10.5194/amt-12-4993-2019) find PSD shape is an important factor in the triple-frequency radar signature (and by extension in the DWR measurements).
A: We agree with the reviewer that our assumption of an inverse exponential PSD might lead to an underestimation of the D0 mentioned in table 2. We have added a discussion of the possible underestimation and dependency of D0 on the PSD shape to the manuscript.
New Text Line 400-403: *As is shown in Mason et al. (2019), the shape of the PSD influences the shape of the triple-frequency signatures, and by extension also the DWR measurements. A narrow PSD with a large D0 might account for the same DWRKaW as a more wide PSD with a smaller D0.*

9. L380: "temperature," → "temperature"
A: Adjusted in the manuscript

10. Fig. 4b: Can you comment on the negative DWR values above the DGL? Can calibration uncertainty (i.e., Section 2.2.3) attribute to this?
A: Yes, calibration uncertainties are the main reason for those slightly negative DWRs. In order to obtain a reliable estimate of the time varying DWR offset due to attenuation and other effects (wet antenna/radome) we need to use a relatively large part of the cloud and accept also Ze values up to -10dBz. This might not perfectly restrict the regions to pure Rayleigh particles. Some remaining differential scattering could produce an overestimation of the DWR which leads to an over-compensation of the W band. The effect is less visible in the stronger DWR classes where the main signal comes from larger ice and snow particles.

11. L398: Specify "more negative" in addition to slightly larger for clarity.
A: Changed in the manuscript
New Text Line 433: *Unlike Ze and DWR-KaW, the MDV are only slightly more negative for the larger DWR-KaW classes*

12. L400–401: Add negative signs to your MDV values for consistency.
A: Added

13. L405: I'm unsure what you mean by "upwind" as it relates to large scale lifting. Can you clarify?
A: We intended to say updraft, not upwind. We have changed that in the manuscript

14. L429: Is this difference between the slow-down on the slow and fast edges in Fig. 5a statistically significant, or at least greater than potential uncertainties in the MDV measurements?
A: In order to address the reviewer's concern, we performed a Kolmogorov-Smirnoff 2-sample test, testing for the null-hypothesis "The velocity distributions at the temperature level where we find the first onset of a slow-down (-18°C for the slow and -16°C for the fast edge) and the temperature level with the largest observed slow down (-14°C for both edges) are equal." At the slow edge, we found that, a p-value of $6.125*10^{-11}$. This p-value is small enough to savely reject the null-hypothesis in favour of the alternative hypothesis that both distributions are significantly different.
For the fast edge we find a p-value of 0.5, suggesting that the null-hypothesis can not be rejected. We believe that the non-significance of the fast-edge slow-down is a result of imperfect filtering of the data. For example, the updraft caused by latent heat released by depositional growth depends on the thermodynamic profile of the atmosphere. Still, we think that the fact that the fast edge velocity decreases consistently between -16 and -14°C is a strong indication for a dynamical feature. It is hard to imagine any microphysical process which could cause such a

phenomenon. Continuous particle growth should increase MDV or at least cause a constant MDV profile.

We have added a paragraph discussing the significance level of the slow down on both edges to section 4.2.

New Text line 480-485: *A Kolmogorov-Smirnoff two-sample test revealed that the slow-down on the slow falling edge is statistically significant, while it is not significant for the fast edge. In case of no updraft, we would expect the fast edge velocity to continuously increase, similarly as for temperatures colder than –16 ° C. So even if the Kolmogorov-Smirnoff test indicates that the slow-down on the fast edge is not statistically significant, we argue that the persistent stagnation of the fall velocities over the temperature range from between –16 ° C and –14 ° C strongly points towards the presence of an updraft.*

15. L456–457: True, but is the case study representative of the entirety of the project as it relates to ZDR?

A: The reference to the case study is only meant as an illustration of the fact that high-ZDR particles and low-ZDR particles are well seperated in the spectrum. This effect can be found in almost all cases analyzed.

16. L471: "confirmed by" → "consistent with"

A: Adjusted in the manuscript (see line 519 in revised document)

17. L472–473: Remove "apparently", "DGL is correlated" → "DGL appears to be correlated"

A: Manuscript adjusted accordingly (see line 520-521 in revised document)

18. L483: The study of Moisseev et al. (2015) is briefly mentioned in Appendix C, but it could be good to relate their findings to what's discussed in this paragraph.

A: In our opinion, it is difficult to relate our statistical findings to the results of single case studies, because usually only large KDP and ZDR cases are selected, while in our case we do not cherry-pick specific case studies were we see large KDP and ZDR but focus on all cases that we observed during the three month campaign. The case study that we showed in Section 3 was meant to illustrate the similarities and differences we see in a case with large KDP and ZDR compared to other studies. In the statistics, our KDP and ZDR (sZDRmax) are smaller than in the case study and also in case studies from previous studies because we also include cases where we do not see large KDP or ZDR occurring at the same time as enhanced DWR. This reduces the magnitude of the median.

19. L498: I think a word was left out for "until the -5°C is reached". Possibly rephrase?

A: We removed the "the" in front of -5°C

New Text line 564-565: *Most notably, signatures related to crystal growth or aggregation that evolved in the DGL appear to persist to lower layers until -5°C is reached.*

20. L506: Parentheses should be like "Takahashi (2014)".
A: Adjusted in the manuscript

21. L510, 642, and possibly elsewhere: You should also acknowledge ice supersaturation as an important factor for the ice crystal habit, as is mentioned in their study.
A: We have added that to the manuscript
New Text (line 581-584): *As mentioned in (Bailey and Hallett, 2009, among others), the shape of the particles does not only depend on temperature, but also on the supersaturation that the particle experiences during growth. During the TRIPEx-pol campaign we do not have sufficient relative humidity information. In the following we therefore only focus our interpretation on the observed temperature-dependent growth of ice particles.*

22. L516: "then" → "than"
A: Adjusted in the manuscript

23. L535–543: Have you looked at or considered other (i.e., mesoscale) effects? You mentioned earlier that many of these events were frontal driven, so it's plausible that frontogenesis or weak/elevated instability can contribute to regions of vertical motion in addition to latent heating by depositional growth.
A: We have not looked at other effects. Since the campaign took place in early to mid-winter, the height of the -15°C isotherm above the ground and also in respect to the height within the cloud changed significantly from case to case. We therefore excluded mesoscale effects in our discussion because we could not think of a reason why such a large-scale driven updraft or weak instabilities should have such a strong temperature dependency. Of course mesoscale effects can contribute to the updraft found at -15°C, but we think that statistically speaking this should be distributed over a larger temperature regime and thus the effect on the median profiles should be small.

24. L548: I agree that most bulk schemes lack the ability to resolve these growth rates and latent heating properly. Maybe add that bin schemes e.g., Lee & Baik 2018; https://doi.org/10.3390/atmos9120475) produce greater latent heat via deposition that leads to stronger updrafts.
A: Thanks for noting this study, we added the reference to the manuscript
New Text (line 624-627): *However, when simulating a heavy rainfall case, Lee and Baik (2018) found that simulations with a bin microphysics schemes reveal intense latent heat release due to depositional growth. This latent heat release is sufficient to*

*cause an updraft and a positive feedback mechanism. The latent heat release in bulk schemes was found to be substantially weaker.*

25. L560 and Appendix C: Have you examined the per-particle KDP for other habits in the scattering database? Since mid-latitude winter systems typically consist of many habits in a radar gate, the sensitivity of KDP by habit as shown in Fig. C1c may mean that the 1/3 contribution of aggregates to the KDP signal may have large uncertainties.

A: So far we have not examined other crystal habits in the KDP estimation. We agree that this estimation has large uncertainty. Our main goal of this ad hoc calculation was to demonstrate that the contribution of aggregates to KDP cannot be neglected, which makes the interpretation of KDP more difficult. In the near future we want to investigate the observed KDP and sZDRmax values further using Monte-Carlo particle modelling with implemented habit prediction.

26. L587: DeMott et al. (2010; https://doi.org/10.1073/pnas.0910818107) also has comprehensive (global) INP measurements.
A: Reference was added (see line 670 in revised document)

27. Conclusions: I also think you should acknowledge that future studies employing in situ microphysical measurements could be a unique opportunity to validate the findings or confirm the speculated processes presented in this study.
A: Agreed, also further laboratory or modelling studies might be useful to validate the findings and hypothesised processes. We have added that to the conclusions.
New Text line 770-771: *Such laboratory studies in addition to in-situ measurements or Monte-Carlo modelling studies could also provide unique opportunities to validate our findings and the hypothesized ice microphysical processes of this study.*

**Answers to reviewer 2**

**General comments:**

One improvement that should benefit this manuscript is using more quantitative methods or language to illustrate differences between radar profiles that are grouped by the aggregate size class or cloud top temperature. For example, differences between the median profiles of a given radar variable from two different classes could be described relative to the standard deviations of the profiles within each class. Or, for example, the lower quantile of one class exceeds the upper quantile of another class at a certain height (or temperature).

A: We agree with the reviewer that we have neglected the description of the distributions/quantiles, and we have included a more detailed description of the quantiles especially in section 4 and 5.2.

As the descriptions in section 4 are currently presented, it is unclear how robust the presumed relations between the radar profiles and these classes (aggregate size and cloud top temperature) are without knowing whether they could be explained simply by random variability from subsetting the data.

A: In order to test the robustness of our statistics, we performed a bootstrapping analysis. We therefore randomly subsampled our dataset into 50% portions 50 times and analysed the distribution of medians of the different subsamples. The shaded areas in Figure 5 represent the 25 and 75 percentile of the distribution of medians if we subsample our dataset and then classify it into CTT classes. The solid line is the median of the entire dataset as our best estimate for the median of the medians. The spread within the 25 and 75 percentiles is low. We therefore think that the medians presented in the paper are made out of robust statistics. We did the same analysis on the DWR-classes statistics (Figure 6). The results are similar: the spread between the 25 and 75 percentile is small. Hence, the statistical analysis of the DWR-classes appears to be robust. We therefore conclude that the differences in radar variable profiles that we see between the classes is beyond what we would expect from randomly grouping into different classes.

Adding more quantitative language to these areas of the manuscript (especially section 5.2) should help better qualify whether the relations are physical or incidental.

A: We agree and changed section 5.2 accordingly (see also answer to first comment)

I also think that the evaluation of the aggregate contribution to KDP requires more discussion of the impact of particle orientation. The single-particle calculations of KDP shown in Appendix C are acceptable for the purposes of illustrating the scattering behavior with respect to size, but likely overestimate the KDP of natural aggregates if a fixed horizontal orientation is used rather than a distribution that accounts for flutter or tumbling. As such, the claims in section 5.1 regarding the aggregate contribution to KDP should also be qualified as highly uncertain.

A: We agree that the calculation and estimated contribution of the aggregates to KDP strongly depends on the aggregate orientation chosen. With this ad hoc calculation we aimed to demonstrate the importance to take aggregates into account when interpreting KDP. It also shows that aggregates alone cannot explain the observed KDP signature even if they are assumed to fall perfectly oriented. In this way, our calculations should be seen as an upper limit estimate of the aggregate contribution to KDP. We have changed the text to underline the uncertainty of our method and that the ⅓ contribution is only to be thought of as the upper limit. In future studies we aim to estimate this in more detail by undertaking studies using Monte-Carlo lagrangian particle model simulations. By doing so, we do not need to explicitly assume a PSD and also different orientations can be accounted for.

**Specific comments:**
Line 102: I believe this relation should be reversed; the Rayleigh regime is valid for particles with size much smaller than the wavelength.
A: agreed, fixed in the manuscript (see Line 102 in revised document)

Lines 180-183: Is the spectral mask simply the region outside of the spectral edges determined by the bins exceeding the noise floor? Please clarify.
A: Yes, with the spectral mask we mask all areas which are outside of the estimated spectral edges from the Ka-Band radar. We have adjusted the text to make it more clear.
New text Line 183: *Our spectral mask is defined by the Doppler velocity bins identified by this method to contain real signal*

Line 252: Please add what specific measurements this correlation refers to.
A: The correlation always refers to the reflectivities of the reference radar (Ka) and one of the other two radars it is compared to (X or W).
New Text Line 253-254: *Further, regions for which the variance of the DWRs exceeds 2 dB^2,* or w*here the correlation between Zes from the reference radar (Ka) and one of the other radars (X, W) is less than 0.7 are discarded*

Line 268: What is the maximum range used for the W-band measurements taken at

30-degree elevation angle? Please add what the horizontal distance is between data at this maximum range and the location of the vertically pointing radars.

A: The maximum range of the polarimetric W-Band radar is 16km ( noted in Table 1). At an elevation angle of 30° this range corresponds to a maximum height above ground of 8km. The maximum horizontal distance at maximum range between the polarimetric radar and zenith radars is 13.86km. We have added a short text in the manuscript.

New Text Line 269-271: *At a maximum range of the polarimetric radar of 16 km (see also Table 1), the height above ground is 8 km and the maximum horizontal distance between the vertically pointing radar and the polarimetric radar is cos(30°) · 16 km = 13.86 km*

Line 305: I think using "slower than" is a bit more confusing than "greater than." Maybe if there is a mention in the text of negative MDV corresponding to motion towards the ground, comparisons to specific values of MDV would be more appropriate than indirectly referring to the absolute value of MDV.

A: We agree and have adjusted the manuscript accordingly. We have also added a short explanation of the magnitude of MDV (and other velocity related variables such as spectral edge velocities)

New Text Line 317-319: *The MDV (Figure 2b) throughout the case are found to be larger than −1.5 m s−1 which indicates unrimed or only slightly rimed particles (Kneifel and Moisseev, 2020). Here we use the convention that negative (MDV) velocities correspond to motion towards the ground. Faster falling particle therefore have smaller (more negative) values than slower falling particles*

Lines 313-315: Please add that the ZDR at an elevation angle of 30 degrees will always be less than that measured at side incidence. It's important to mention this difference because other studies of ice microphysics often observe ZDR at elevation angles closer to side incidence (i.e., < 5 degrees).

A: We agree and mention it now explicitly in the manuscript.

New Text Line 332: *Note that at 30° elevation ZDR is expected to be in general smaller than ZDR measured at lower elevation angles which have often been used in previous studies where data from lower frequency scanning radar systems have been analysed*

Line 338: It is preferable to say that the spectrum shifts rightward or towards larger values since there are weakly positive velocities in Fig. 3e near -15 degrees C.

A: Agreed. We changed the manuscript accordingly (see adjustment in Line 359 in revised manuscript)

Line 340-341: The wording here is a bit unclear. Do the authors mean something like: the main mode contributes more power to the spectrum than the secondary

mode and therefore shifts of the main mode with respect to Doppler velocity dominate changes in MDV with height? Please clarify.

A: Yes this is what we were trying to say. We have reworded the phrase and hope it is clearer now!

New Text Line 361-363: *The main mode contributes more power to the spectrum than the newly formed secondary mode. Therefore, shifts of the main mode towards slower or faster velocities dominate changes in MDV. Hence, the slow-down of the main spectral mode at −12.5 ° C reduces the MDV at this temperature.*

Line 342: At -12 degrees C?

A: Yes, roughly at -12°C

Line 345: How much of these oscillations in KDP are due to noise in the PhiDP profiles compared to a microphysical signal?

A: We have investigated the variability of KDP due to the noise in PhiDP by analysing PPI scans obtained at 85° elevations. At such high elevations, KDP is expected to be close to 0, deviations from 0 would correspond to the uncertainty due to noise in PhiDP. We applied the same processing to the PPI scans that we used on the 30° elevation data. However, we were only able to average over one PPI scan which lasted 1 minute in contrast to the 5 minute average we applied to the 30° elevation observations. Figure 4 shows the statistical analysis of KDP obtained from these PPI scans. The variability of KDP is rather small. For most temperature regions the median KDP oscillates between -0.05 and 0.01. Even the quantiles are within -0.15 and 0.1°/km. Since we were only able to apply a 1 minute average to the KDP from the PPI scans (as opposed to the 5 min average we apply on the 30° elevation data), we expect the variability of KDP at 30° elevation to be even less.

[Figure]

Figure 4: median KDP (solid line) and quantiles (shading) of KDP calculated from PPI at 85° elevation

Lines 351-359: I largely agree with this assessment. However, the lack of layered KDP enhancements observed in this study may also be due to a lack of strong forcing associated with mesoscale snowbands. In these snowband cases, there may be more particles and/or more rapid dendritic growth leading to more intense aggregation and thus a more rapid depletion of pristine ice crystals. The higher sensitivity radars used in this project would be able to detect weaker vapor growth and aggregation cases where the ice crystal depletion is slower, extending the KDP enhancement farther down. KDP observations of these weaker cases at S-band or X-band would likely show near-zero KDP throughout the profile. So in order for the low-frequency radars to detect measurable KDP enhancements in snow, there may need to be more substantial vapor growth and subsequent aggregation. Please address this potential for selection bias with respect to low-frequency radars in the text.

A: We have extended our discussion of this potential selection bias and included your suggestions.

New Text Line 380-386: *Another reason for the less layered appearance of KDP and ZDR might be an under-representation of cases with strong forcing conditions during TRIPEx-pol. More intense vertical air motions are expected to result in a larger*

*concentration of particles and abundance of dendrites that is expected to lead to stronger aggregation and more intense depletion of ice crystals (e.g. Moisseev et al., 2015; Schrom et al., 2015). Radars operating at longer wavelengths can clearly detect these cases with large concentrations of ice crystals at –15 ° C but might miss cases with weaker forcing due to sensitivity limits. The differences in sensitivity between W-Band and lower frequency radars might cause a selection bias of low-frequency radars with respect to stronger depositional growth and aggregation cases.*

Line 369: Does continuous here refer to profiles without any masked regions?

A: Continuous in this case refers to profiles where there is a signal in all range gates between -10 and -20°C. With this we want to avoid multi-layered cases and therefore weak signals due to sublimation or new cloud tops. Usually this does not refer to masked regions, because during the DWR-calibration, entire profiles were discarded if the mentioned criteria were not met.

Line 388: Please use "dB" instead of "dBZ" for differences in reflectivity values.

A: Adjusted in the manuscript

Lines 396-397: Please change to "magnitude of MDV increases."

A: Adjusted in the manuscript (see line 429-430 in revised document)

Line 405: Please change "upwind" to "updraft."

A: Adjusted in the manuscript (see line 441 in revised document)

Line 422: How are the slow and fast edges of the spectra determined? Are they the first and last bins above the noise threshold?

A: The slow and fast edges were determined as described in section 2.2.1. We take the first and last bin 3dB above the noise threshold. In cases of strong Ze signals, there might be spectral leakages that could artificially broaden the spectra. We therefore further neglect all spectral lines which are lower than 40dBz with respect to the maximum spectral line.

Lines 426-427: Does this assumption that new particles decrease the slow edge of the spectrum require that these particles have a minimum fall velocity? For example, if the new particles only become detectable once their fall speed is 0.5 m/s, isn't it possible that they would have no effect on the slow edge velocity?

A: From our practical experience we see new ice particle modes (can be distinguished from liquid drops by using for example sLDR) typically occurring at 0.2-0.3 m/s. This is also true for temperature regions of preferential columnar or needle growth such as close to the -7°C level. In this region, one would expect the particles to fall fastest even at small sizes due to their small cross sectional area. Rough estimates from scattering computations indicate (assuming typical range of concentrations) that ice crystals exceeding 100um in maximum size are well detectable by common cloud radars. This size range roughly matches the 0.2-0.3

m/s Doppler velocities where the first signal is usually detected (of course large variabilities in terminal velocities exist for individual crystals). Certainly, we cannot completely exclude the scenario which the reviewer described but we consider it as very unlikely. We are quite confident that new ice particle formation leads to a substantial decrease of the slow edge velocity especially if analysing a large dataset as presented in our study.

Line 499: Given that the example case study seems to have much higher skewness, KDP, and maximum spectral ZDR (near 4 dB according to Fig. 3f) compared to the bulk statistics, there should be some brief discussion of the uniqueness of that case relative to the others in the dataset.
A: The example profiles shown in Fig 3 have been selected to show the general behaviour in different radar variables in a more pronounced way as visible in the bulk statistics. The case study itself represents an event with stronger signals but it is certainly not an exceptional case. We mention this now also in the text.

New Text Line 305-307: *The case study selected shows more pronounced signals then the average profiles discussed later in the statistical analysis. However, the case is not an exceptional event and similar profiles of radar variables can be found frequently at JOYCE-CF during similar winter cases*
New Text Line 555-562: *Comparing the results of the statistical analysis to the case study presented in Section 3, we note the smaller values of KDP, sZDRmax and skewness found in the statistics. In the case study, sZDRmax values of up to 4 dB at around −15 ° C were reached, alongside a maximum KDP of approx. 2 ° km−1 and a skewness of 1.3. However, in the statistics we classify the profiles with the maximum DWRKaW. In Figure 2, one can see that we do not always find an increase in KDP or sZDRmax for increasing DWRKaW. For example, at 06:30 UTC, a strong increase in KDP below −15 ° C coincides with an enhanced sZDRmax and DWRKaW. At later periods of the day, for example at 18:00 UTC, we see enhanced DWRKaW without enhanced KDP and sZDRmax. Those examples explain why the medians in our statistical analysis are shifted to smaller values.*

Line 525: ZDR also tends to saturate as dendritic growth occurs because of the generally decreasing effective density of the particles with size. Please add some mention of this effect.
A: We added that to the manuscript
New Text Line 600: *Further, in case of dendritic growth, the effective density of the particle decreases with size, leading to a saturation of ZDR*

Line 558: The uncertainty in this value for the aggregate KDP contribution needs to be more clearly stated. For example, the orientation behavior of the aggregates can have a large impact on the measured KDP.

A: See also our answer to general comment 2. In general, for a given elevation angle, the strongest polarimetric signatures (both ZDR and KDP) are produced when particles are perfectly alligned either horizontally or vertically. Our assumption of perfect horizontal alignment of all particles is therefore an upper limit estimate for the possible contribution of aggregates to the total KDP. Certainly, the more realistic assumption of aggregate tumbling during their fall will reduce their contribution to KDP. We have added a sentence noting that to the appendix.

New Text Line 851-853: *This high contribution of aggregates to the observed KDP is most likely an upper limit, since we assume the aggregates to be perfectly horizontally aligned. The naturally occurring tumbling and fluttering of the particles within clouds would reduce the KDP (and ZDR) produced by aggregates.*

Lines 561-562: Please add a caveat here that there may be non-Rayleigh effects on KDP at larger size parameters than those examined in this study.

A: added to the manuscript

New Text Line 641-642: *It should be ntoed that this behaviour is expected to change for increasing size parameters (e.g., at radar frequencies higher than W-Band)*

Lines 612-613: Please reword this sentence for clarity.

A: We have adjusted this section to include more quantitative language and discussion of the percentiles. The sentences in this line were changed as well.

Lines 612-616: The comparisons between the radar variable profiles with different cloud top temperatures need to be more carefully stated in terms of how significant they are relative to sampling errors between the different groups. In other words, are the differences in particular radar variable profiles beyond what would be expected from randomly grouping the profiles into different classes?

A: In order to test the robustness of our statistics, we performed a bootstrapping analysis. We therefore randomly subsampled our dataset into 50% portions 50 times and analysed the distribution of medians of the different subsamples. This is also addressed in our answer to general comment 2. We concluded that the differences in radar variable profiles that we see between the classes is beyond what we would expect from randomly grouping into different classes. See also Figures 5 and 6 below.

[Figure]

Figure 5: bootstrapping analysis of the CTT-statistics: distribution of medians of 50 subsamples. The subsample size was 50% of the original dataset. The shaded areas indicate the 25 and 75 percentile of the distribution of medians. The solid line is the median of the original dataset.

[Figure]

Figure 6: bootstrapping analysis of the DWR-statistics: distribution of medians of 50 subsamples. The subsample size was 50% of the original dataset. The shaded areas indicate the 25 and 75 percentile of the distribution of medians. The solid line is the median of the original dataset.

Line 654: Please clarify what properties of the sedimenting particles are being considered to have no effect on KDP and sZDRmax here.
A: We are not sure if we understand the reviewer here. In this part, we are solely describing the evolution of the observed profiles: "Similar as for KDP, no strong difference is found in sZDR$_{max}$ for particles sedimenting into the DGL from above."

Lines 752-754: What orientation assumptions are being used to calculate the aggregate KDP?
A: We have used horizontally aligned particles. (Already stated in line 713)

Does the simulated ZDR for the aggregates with these PSD assumptions agree with the measurements?

A: We obtain a ZDR of 2.5dB for the perfectly oriented aggregates assumed in our example calculation. This value is certainly larger than the 1.25dB observed at -12°C. sZDRmax in the statistics is quite small. Since there are cases with very small sZDRmax values, the median of our statistics is shifted towards smaller values. Also, fluttering would reduce the ZDR of the aggregates.

---

## Author Response (AR2)

**Answers to reviewers, 2nd revision**

We would like to thank the reviewers again for their time and effort reviewing this manuscript and providing helpful comments and suggestions. The comments from the reviewers are marked in black, our answers in red and new text added to the manuscript in blue. In the revised document, all new text is marked in blue, and deleted text is crossed out in red.

**Report #1:**

1. L292: The addition of this section is helpful to assess the accuracy of these Cloudnet profiles. However, computing the mean temperature difference between the radiosondes and Cloudnet can be misleading if Cloudnet exhibits a cool bias in one layer and a warm bias in another. Wouldn't RMSD or some other metric better account for these differences?

A: In Dias-Neto 2021, also the RMSD was calculated for the comparison between the Cloudnet temperature information and the temperature measured by the radiosondes. Dias-Neto 2021 found that the RMSD=1.1°C. Further, the error is distributed around 0°C for temperatures between -20 and 20°C; for colder temperatures Cloudnet overestimates the temperature. We added the following sentence to Line 293: Further, the root-mean square difference was found to be RMSD=1.1°C.

2. L637–L650: Although it is true that SIP may augment the concentration of ice crystals, are the authors suggesting that mechanisms such as ice fragmentation were so prevalent for this study that it exceeds any impact that aggregation typically has at reducing the concentration of KDP-producing crystals (L628)? Couldn't horizontal/vertical winds, some sort of lifting mechanism, instability, etc. also play a role?

A: Our main goal in this paragraph is to provide a possible explanation for our finding that the increase of KDP towards the bottom of the DGL is correlated with DWR class. At first, this appears to be counter intuitive as aggregation should reduce the number of particles. However, as aggregation is connected to collision of particles, it would be in our opinion quite logical that more collisions could also lead to more fragments. We absolutely agree with the reviewer that other mechanism might be also very relevant for the vertical evolution of number concentration but we somehow doubt that for example lifting or instability would show such a systematic temperature dependence as we find it in our statistics.

**Report #2:**

Line 630-631: I still think that mentioning this factor of three ratio of dendritic to aggregate KDP requires more qualifications regarding the variability in dendritic and aggregate shapes. This shape variability produces a range of KDP values for each particle type, increasing uncertainty in the value of the ratio.

In order to address the reviewer's concern about our example calculation, we repeated the experiment using the scattering properties of the other aggregate types available in the Lu database (see Figure 1). As expected, there is a significant difference, depending on the density and type of monomer assumed. The aggregates with the lowest density (LDt-P1d) produce the smallest Ze, ZDR and KDP. Assuming these low-density particles, in order to match the DWR of 4.2dB,  $\Lambda$ was estimated to be 1.3\*10^-3. For this  $\Lambda$ , N0 of 65\*10^5 has to be assumed to match Ze of 10.2dB. For this PSD and particle type LDt-P1d, a KDP of 1.9 was estimated. So with this extreme, low-density particle type, we would obtain a KDP which even exceeds the observed KDP. In fact, by changing our particle type to low density aggregates the KDP contribution by aggregates would even be more important to be taken into account. However, we would like to stress again that this calculation experiment was not meant to provide final conclusions on the KDP contribution of aggregates. It rather is intended to demonstrate that the KDP contribution can not be simply neglected and that previously calculated low KDP of aggregates might be mainly caused by the used unrealistic scattering model (low density and refractive index connected to effective medium approximation and spheroidal shape approximation).

Figure 1: scattering properties of all aggregate types available in the Lu et al. (2016) database.